# Bridging the Spectrum Gap: Mid-Frequency Augmentation and Key-Frequency Mining for Multivariate Time Series

## Abstract

Recent advancements have progressively incorporated frequency-based techniques into deep learning models, leading to notable improvements in accuracy and efficiency for time series analysis tasks. However, the **Mid-Frequency Spectrum Gap** in the real-world time series, where the energy is concentrated at the low-frequency region while the middle-frequency band is negligible, hinders the ability of existing deep learning models to extract the crucial frequency information. Additionally, the shared **Key-Frequency** in multivariate time series, where different time series share indistinguishable frequency patterns, is rarely exploited by existing literature. This work bridges these two gaps by: *(i)* introducing a novel module, 'Adaptive Mid-Frequency Energy Optimizer', based on convolution and residual learning, to emphasize the significance of mid-frequency bands; *(ii)* proposing an 'Energy-based Key-Frequency Picking Block' to capture shared Key-Frequency, which achieves superior inter-series modeling performance with fewer parameters; *(iii)* employing 'Key-Frequency Enhanced Training' strategy to further enhance Key-Frequency modeling, where spectral information from other channels is randomly introduced into each channel. Our approach advanced multivariate time series forecasting on the challenging Traffic, ECL, and Solar benchmarks, reducing MSE by 4%, 6%, and 5% compared to the previous SOTA iTransformer. Code is available at this **Anonymous Repo**: https://anonymous.4open.science/r/ReFocus-2889.

## 1 Introduction

Accurate forecasting of time series offers reference for decision-making across various domains (Lim & Zohren, 2021; Torres et al., 2021), including weather (Du et al., 2023), economics (Oreshkin et al., 2020), and energy (Dong et al., 2023; Liu et al., 2022b). Especially, long-term multivariate time series forecasting (LMTSF) emerges as a prominent area of interest in academic research (Wang et al., 2024d; Wen et al., 2022) and industrial applications (Cirstea et al., 2022), offering the advantage of capturing complex interdependencies and trends across multiple variables.

Recently, the powerful representation capabilities of neural networks, such as Multi-Layer perception (MLPs) (Yi et al., 2023c; Han et al., 2024), Transformers (Zhou et al., 2022c; Nie et al., 2023), and Temporal Convolution Network (TCNs) (Eldele et al., 2024; Liu et al., 2022a), have significantly advanced deep learning-based LMTSF. These approaches can be broadly categorized into two/three folds: time-domain-based (Han et al., 2024; Nie et al., 2023; Liu et al., 2022a) and frequency-domain-based (Yi et al., 2023c; Zhou et al., 2022c; Eldele et al., 2024) methods, or mixed time & frequency. Time-domain methods are intuitive, handling nonlinearity and non-periodic signals directly from the raw sequence (Li et al., 2023) using Transformers (Zhou et al., 2022a), TCN (Donghao & Xue, 2024), or MLP (Wang et al., 2024a). The latest study Yi et al. (2024) highlights that time-domain forecasters face challenges such as vulnerability to high-frequency noise, and computational inefficiencies. While frequency-domain-based methods usually transform the time-domain data to the frequency spectrum by Fast Fourier transform (FFT) (Yi et al., 2023a). Then other operations (Self-attention (Zhou et al., 2022c), Linear mapping (Xu et al., 2024b; Yi et al., 2023c), etc.) are employed to extract frequency information. These methods benefit from advantages such as computational efficiency (Fan et al.,

2024; Xu et al., 2024b), periodic patterns extracting (Wu et al., 2023; Dai et al., 2024), and energy compaction (Yi et al., 2023c;b).

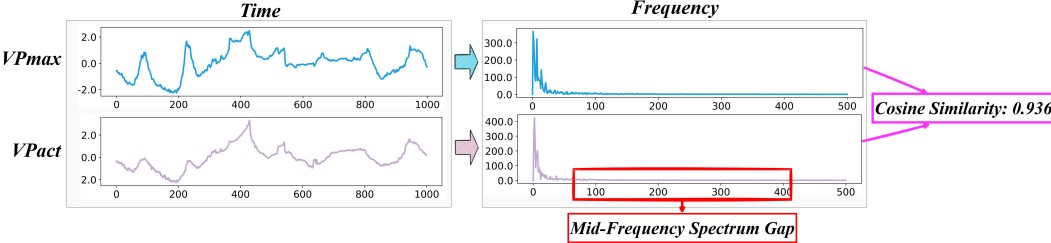

Figure 1: The **Mid-Frequency Spectrum Gap** and the shared **Key-Frequency** (high similarity in frequency spectra across variables) on Weather dataset. VPmax means 'Maximum Vapor Pressure' and VPact means 'Actual Vapor Pressure'.

However, existing frequency-domain-based forecasters usually face TWO significant challenges when dealing with real-world long-term time series: the **Mid-Frequency Spectrum Gap** and the shared **Key-Frequency modeling**.

- **Mid-Frequency Spectrum Gap** (Figure 1 Red box) refers to a condition where the energy of the spectrum is concentrated in the low-frequency regions, resulting in the mid-frequency band being negligible. Low-frequency components capture long-term trends, often contributing to mean shifts when overly concentrated (Stock & Watson, 2002; Granger & Newbold, 1974; Chatfield & Xing, 2019). So this Mid-Frequency Spectrum Gap will introduce **Nonstationarity** (Cheng et al., 2015; Liu et al., 2022c), where the mean and variance of time series change over time, and make time series less predictable. Furthermore, such uneven energy distribution challenges existing deep-learning models to extract critical patterns (Tishby & Zaslavsky, 2015; Xu et al., 2024a; Rahaman et al., 2019). So, addressing this Mid-Frequency Spectrum Gap is crucial for enhancing the feature extraction capabilities of deep learning-based forecasters (Park et al., 2019; Bai et al., 2018; Guo et al., 2019). Currently, widely used methods for processing spectra, such as Filters (Asselin, 1972), and **RevIN** (Kim et al., 2022; Liu et al., 2022c)—a technique previously applied to address nonstationarity—are not effective in resolving this issue. Conversely, convolution with residual connections has effectively handled spectral information (Can & Timofte, 2018; Chakraborty & Trehan, 2021), providing a potential solution.

- Meanwhile, the second challenge: the shared **Key-Frequency Modeling** (Figure 1 Pink box) has the disadvantage that distinct time series can exhibit indistinguishable frequency patterns, potentially leading to challenges in accurately differentiating and analyzing individual series within a multivariate context (Yu et al., 2023; Piao et al., 2024). However, existing approaches have largely overlooked this critical characteristic. Meanwhile, energy, which is the square of the amplitude of the spectrum, is proven as an effective tool for identifying certain frequency patterns in the multivariate case (Bógalo et al., 2024; Chekroun & Kondrashov, 2017; Sundararajan & Bruce, 2023).

Based on the above observations, this work mainly addresses two critical questions: (1) How can the Mid-Frequency Spectrum Gap be resolved to achieve a more evenly dispersed spectrum? (2) How can inter-series dependencies be efficiently modeled by leveraging the shared Key-Frequency? To tackle challenge 1, we propose the 'Adaptive Mid-Frequency Energy Optimizer' (AMEO), a convolution- and residual learning-based solution. It adaptively scales the frequency spectrum by assigning higher scaling factors to lower frequencies, thereby dispersing the spectrum. To address challenge 2, for the second challenge, we introduce the 'Energy-based Key-Frequency Picking Block (EKPB)', which features fewer parameters and faster inference speeds compared to the Transformer Encoder (Liu et al., 2024b) and MLP-Mixer (Chen et al., 2023). EKPB extracts shared frequency information across channels effectively. We also propose a 'Key-Frequency Enhanced Training' strategy (KET) which incorporates spectral information from other channels during training to enhance extraction of shared Key-Frequency that may not be included in the training set. We name our full framework **ReFocus**, (**Re**sidual **F**requency **O**ptimization and **C**ross-channel **U**nified **S**pectrum modeling), as it refocuses model capacity on underutilized yet critical spectral regions and inter-series frequency.

Our contributions are summarized as follows:

- We theoretically and empirically demonstrate that existing RevIN and high/low-pass filters fail to address the Mid-Frequency Spectrum Gap. We propose AMEO, a novel approach based on convolution and residual learning that significantly enhances mid-frequency feature extraction.
- We propose EKPB to capture shared Key-Frequency across channels, which achieves superior inter-series modeling capacity with lower parameters.
- We propose KET, where spectral information from other channels is randomly introduced into each channel, to enhance the extraction of the shared Key-Frequency.
- Our approach outperforms the previous SOTA iTransformer by reducing MSE by 4%, 6%, and 5% on the challenging Traffic, ECL, and Solar datasets, respectively, establishing new benchmarks in multivariate time series forecasting.

## 2 RELATED WORK

**Advancement in Recent Deep Learning-based Time Series Forecasting**     Recent advancements in deep learning-based time series forecasting can be broadly categorized into three key areas: (1) the application of sequential models to time series data, (2) the tokenization of time series, and (3) the exploration of intrinsic patterns within time series. Efforts in the first area have focused on deploying various architectures for time series forecasting, including Transformer (Wu et al., 2021; Wang et al., 2024b), Mamba (Ahamed & Cheng, 2024; Wang et al., 2024e), MLPs (Wang et al., 2024a; Das et al., 2023; Yu et al., 2024a), RNNs (Lin et al., 2023), Graph Neural Networks (Shang et al., 2024), TCNs (Wang et al., 2023), and even Large Language Models (LLMs) (Jin et al., 2024; Liu et al., 2024d;c). The second direction has witnessed groundbreaking developments, particularly in Patch Embedding (Nie et al., 2023) and Variate Embedding (Liu et al., 2024b). The final area explores modeling complex relationships, including the inter-series dependencies (Ng et al., 2022; Chen et al., 2024), the dynamic evolution within a sequence (Du et al., 2023; Zhang et al., 2022), or both (Yu et al., 2024b; Liu et al., 2024a).

**Time Series Modeling with Frequency**     Frequency as a key feature of time series data, has inspired numerous works (Yi et al., 2023a). FITS (Xu et al., 2024b) employs a simple frequency-domain linear, getting results comparable to SOTA models with 10K parameters. Autoformer (Wu et al., 2021) introduces the auto-correlation mechanism, leveraging FFT to improve self-attention. FEDformer (Zhou et al., 2022c) further calculates attention weights from the spectrum of queries and keys. FiLM (Zhou et al., 2022b) applies Fourier analysis to preserve historical information while filtering out noise. FreTS (Yi et al., 2023c) incorporates frequency-domain MLP to model both channel and temporal dependencies. TimesNet (Wu et al., 2023) utilizes FFT to extract periodic patterns. FilterNet (Yi et al., 2024) proposes a filter-based method from the perspective of signal processing.

However, they do not address the Mid-Frequency Spectrum Gap and shared Key-Frequency modeling. In contrast, our method employs 'Adaptive Mid-Frequency Energy Optimizer' to improve mid-frequency feature extraction and introduces 'Energy-based Key-Frequency Picking Block' with 'Key-Frequency Enhanced Training' strategy to capture shared Key-Frequency across channels.

## 3 METHODOLOGY

### 3.1 PROBLEM DEFINITION

Given a multivariate time series input $X \in \mathbb{R}^{C \times T}$, multivariate time series forecasting tasks are designed to predict its future $F$ time steps $Y \in \mathbb{R}^{C \times F}$ using past $T$ steps. $C$ is the number of variates or channels.

### 3.2 PRELIMINARY ANALYSIS

This section presents why RevIN (Kim et al., 2022; Liu et al., 2022c), High-pass, and Low-pass filters fail to address the Mid-Frequency Spectrum Gap. Let the input univariate time series be $x(t)$ with length $T$ and target $y(t)$ with length $F$.

**Definition 3.1** (Frequency Spectral Energy). The Fourier transform of $x(t)$, $X(f)$, and its spectral energy $E_X(f)$ is given by:

$$X(f) = \sum_{t=0}^{T-1} x(t)e^{-i2\pi ft/T-1}, \quad f = 0, 1, \ldots, T-1$$

$$E_X(f) = |X(f)|^2. \tag{1}$$

**Impact of RevIN on Frequency Spectrum**

**Definition 3.2** (Reversible Instance Normalization). Given a **forecast model** $f : \mathbb{R}^T \to \mathbb{R}^F$ that generates a forecast $\hat{y}(t)$ from a given input $x(t)$, RevIN is defined as:

$$\hat{x}(t) = \frac{x(t) - \mu}{\sigma}, \quad t = 0, 1, \ldots, T-1$$

$$\hat{y}(t) = f(\hat{x}(t)), \quad \hat{y}(t)_{rev} = \hat{y}(t) \cdot \sigma + \mu,$$

$$\mu = \frac{1}{T}\sum_{t=0}^{T-1} x(t), \quad \sigma = \sqrt{\frac{1}{T}\sum_{t=0}^{T-1}(x(t) - \mu)^2}. \tag{2}$$

**Theorem 3.3** (Frequency Spectrum after RevIN). *The spectral energy of $\hat{x}(t)$ (transformed using RevIN):*

$$E_{\hat{X}}(0) = 0, \quad f = 0,$$

$$E_{\hat{X}}(f) = \left(\frac{1}{\sigma}\right)^2 |X(f)|^2, \quad f = 1, 2, \ldots, T-1. \tag{3}$$

The proof is in Appendix A.1. Theorem 3.3 suggests that RevIN scales the absolute spectral energy by $\sigma^2$ but does not affect its relative distribution except $E_{\hat{X}}(0) = 0$. Thus, RevIN preserves the relative spectral energy distribution and leaves the Mid-Frequency Spectrum Gap unresolved. *However, our experiments still employ RevIN to ensure a fair comparison with other baselines.*

**Impact of High- and Low-pass filter**    We still define $\hat{x}(t)$ to be the filtered (processed) signal, obtained by applying a filter $H(f)$ (High/Low-pass filter). The filter $H(f)$ is 1 in the passband (High/Low frequency) and 0 in the stopband (Middle frequency). So $E_{\hat{X}}(f) = 0, \quad E_{\hat{X}} \leq E_X(f)$ for middle frequencies, which creates even larger gap.

## 3.3 Overall Structure of The Proposed ReFocus

In this section, we elucidate the overall architecture of **ReFocus**, depicted in Figure 2. We define frequency domain projection as $D1 \to D2$ representing a projection from dimension $D1$ to $D2$ in the frequency domain (Xu et al., 2024b). Initially, we apply **AMEO** to the input $X \in \mathbb{R}^{C \times T}$, yielding the processed spectrum $X_{am} \in \mathbb{R}^{C \times T}$. Next, we use a projection $T \to D$ to transform $X_{am}$ into the Variate Embedding $X_{em} \in \mathbb{R}^{C \times D}$ (Liu et al., 2024b). Then, $X_{em}$ go through $N$ **EKPB** to generate representation $H_{N+1}$, which is projected to obtain final prediction $\hat{Y}$.

**Adaptive Mid-Frequency Energy Optimizer**    Building upon the **Preliminary Analysis**, we propose a convolution- and residual learning-based solution to address the Mid-Frequency Spectrum Gap, which we denoted as AMEO.

**Definition 3.4** (Adaptive Mid-Frequency Energy Optimizer). AMEO is defined as:

$$\hat{x}(t) = x(t) - \frac{\beta}{K}\sum_{k=0}^{K-1} \tilde{x}(t + K - 1 - k),$$

$$\tilde{x}(t) =$$

$$\begin{cases} x(t - (\frac{K}{2} + 1)), & \text{if } \frac{K}{2} + 1 \leq t < T + \frac{K}{2} + 1, \\ 0, & \text{if } 0 \leq t < \frac{K}{2} + 1 \text{ or } T + \frac{K}{2} + 1 \leq t < T + K. \end{cases} \tag{4}$$

It is equivalent to $x = x - \beta \cdot Conv(x)$. $Conv$ is a 1D convolution (Zero-padding at both ends, stride $s = 1$, kernel size $K$, with values initialized as $\frac{1}{K}$). $\beta \in \mathbb{R}^1$ is a hyperparameter.

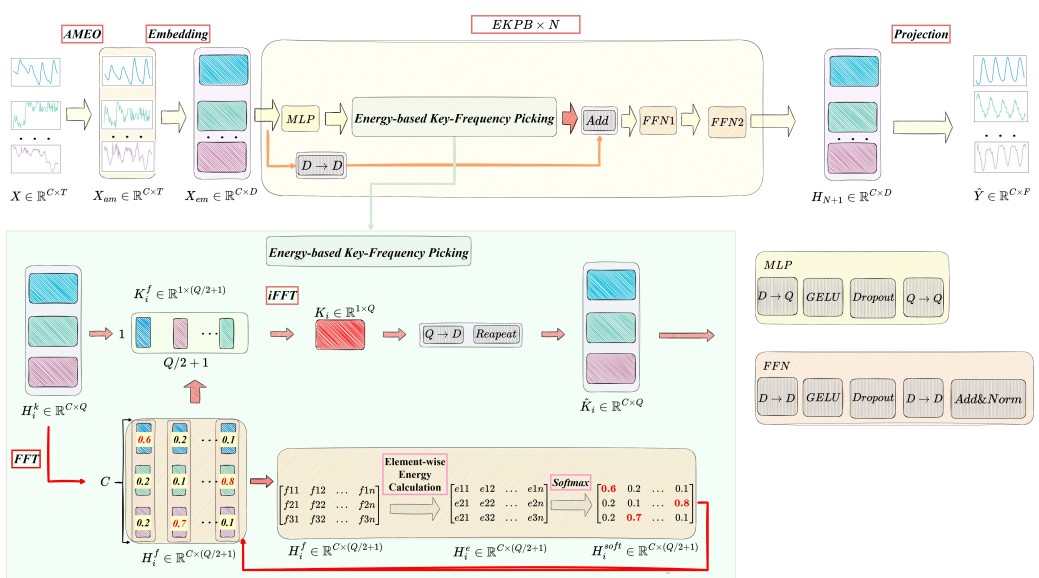

Figure 2: General structure of **ReFocus**. 'Adaptive Mid-Frequency Energy Optimizer (AMEO)' enhances mid-frequency components modeling, and 'Energy-based Key-Frequency Picking Block' (EKPB) effectively captures shared Key-Frequency across channels

**Theorem 3.5** (Frequency Spectrum after AMEO). *The spectral energy of $\hat{x}(t)$ obtained using AMEO:*

$$E_{\hat{X}}(f) = |X(f)|^2 \left\{ 1 - \beta \cdot \underbrace{\frac{1}{K} \sum_{k=0}^{K-1} e^{i2\pi f(\frac{K}{2}-k-2)/T-1}}_{G(f)} \right\}^2 \tag{5}$$

The proof is in Appendix A.2. We have $E_{\hat{X}}(f) = |X(f)|^2 (1 - \beta \cdot G(f))^2$. Generally, $G(f)$ behaves as a decay function, gradually reducing its value from **One** to **Zero**. Such **decay behavior** makes AMEO relatively enhances mid-frequency components, thus addressing the Mid-Frequency Spectrum Gap.

**Energy-based Key-Frequency Picking Block**   In each **EKPB**, the input $H_i \in \mathbb{R}^{C \times D}(H_1 = X_{em})$ is first processed through an MLP to generate $H_i^k \in \mathbb{R}^{C \times Q}$. Then, FFT is applied to get $H_i^f \in \mathbb{R}^{C \times (Q/2+1)}$. For $H_i^f$, we calculate its energy, denoted as $H_i^e \in \mathbb{R}^{C \times (Q/2+1)}$. A cross-channel softmax is then applied to $H_i^e$ per frequency to obtain a probability distribution $H_i^{soft} \in \mathbb{R}^{C \times (Q/2+1)}$. Using $H_i^{soft}$, we select values from $H_i^f$ across channels for each frequency, resulting in $K_i^f \in \mathbb{R}^{1 \times (Q/2+1)}$, which represents the Shared Key-Frequency across all channels. Then iFFT is performed on $K_i^f$ to get $K_i \in \mathbb{R}^{1 \times Q}$, followed by projection $Q \to D$ and repeating (C times) to get $\hat{K}_i \in \mathbb{R}^{C \times D}$. This $\hat{K}_i$ is point-wisely added to $\hat{H}_i \in \mathbb{R}^{C \times D}$, which is the projection of $H_i$ using projection $D \to D$. Then, an MLP and $Add\&Norm$ is applied to the result $HK \in \mathbb{R}^{C \times D}$ to fuse inter-series dependencies information, and another MLP and $Add\&Norm$ is used to capture intra-series variations (Liu et al., 2024b). The output of each **EKPB** is $\hat{O}_i \in \mathbb{R}^{C \times D}$, where $H_{i+1} = \hat{O}_i$.

### 3.4 KEY-FREQUENCY ENHANCED TRAINING STRATEGY

In real-world time series, certain channels often exhibit spectral dependencies, which may not be fully captured in the training set, and the specific channels with such dependencies are also unknown (Geweke, 1984; Zhao & Shen, 2024). So this work borrows insight from recent advancement of mix-up in time series (Zhou et al., 2023; Ansari et al., 2024), randomly introducing spectral

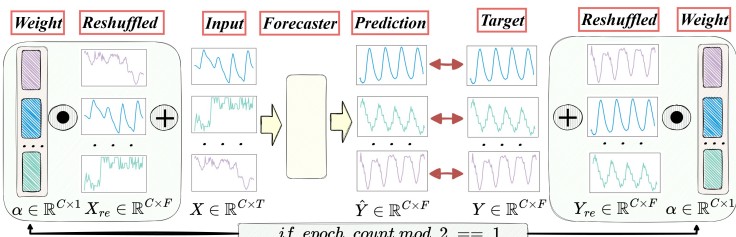

Figure 3: General process of the **Key-Frequency Enhanced Training strategy (KET)**, where spectral information from other channels is randomly introduced into each channel, to enhance the extraction of the shared Key-Frequency.

information from other channels into each channel, to enhance the extraction of the shared Key-Frequency, as in Figure 3. Given a multivariate time series input $X \in \mathbb{R}^{C \times T}$ and its ground-truth $Y \in \mathbb{R}^{C \times F}$, we generate a pseudo sample pair:

$$
\begin{aligned}
X' &= iFFT(FFT(X) + \alpha \cdot FFT(X[\text{perm}, :])), \\
Y' &= iFFT(FFT(Y) + \alpha \cdot FFT(Y[\text{perm}, :])).
\end{aligned}
\tag{6}
$$

$\alpha \in \mathbb{R}^{C \times 1}$ is a weight vector sampled from a normal distribution, perm is a reshuffled channel index. Since $FFT$ and $iFFT$ are linear operations, this mix-up process can be equivalently simplified in the **Time Domain**:

$$
\begin{aligned}
X' &= X + \alpha \cdot X[\text{perm}, :], \\
Y' &= Y + \alpha \cdot Y[\text{perm}, :]
\end{aligned}
\tag{7}
$$

We alternate training between real and synthetic data to preserve the spectral dependencies in real samples. This combines the advantages of data augmentation, such as improved generalization, while mitigating potential drawbacks like over-smoothing and training instability (Ryu et al., 2024; Alkhalifah et al., 2022).

## 4 EXPERIMENTS

### 4.1 EXPERIMENTAL SETTINGS

This section first introduces the whole experiment settings under a fair comparison. Secondly, we illustrate the experiment results by comparing **ReFocus** with the **TEN** well-acknowledged baselines. Further, we conducted an ablation study to comprehensively investigate the effectiveness of the 'Adaptive Mid-Frequency Energy Optimizer' (**AMEO**), 'Energy-based Key-Frequency Picking Block' (**EKPB**), and 'Key-Frequency Enhanced Training strategy' (**KET**).

**Datasets**    We conduct extensive experiments on selected **Eight** widely-used real-world multivariate time series forecasting datasets, including Electricity Transformer Temperature (ETTh1, ETTh2, ETTm1, and ETTm2) (Zhou et al., 2022a), Electricity, Traffic, Weather used by Autoformer (Wu et al., 2021), and Solar_Energy datasets proposed in LSTNet (Lai et al., 2018). For a fair comparison, we follow the same standard protocol (Liu et al., 2024b) and split all forecasting datasets into training, validation, and test sets by the ratio of 6:2:2 for the ETT dataset and 7:1:2 for the other datasets. More can be found in the Appendix.

**Evaluation protocol**    Following TimesNet (Wu et al., 2023), we use Mean Squared Error (MSE) and Mean Absolute Error (MAE) for the evaluation. We follow the same evaluation protocol, where the input length is set as $T = 96$ and the forecasting lengths $F \in \{96, 192, 336, 720\}$. All the experiments are conducted on a single NVIDIA GeForce RTX 4090 with 24G VRAM. The MSE loss function is utilized for model optimization. To foster reproducibility, we make our code, and training scripts available in this **Anonymous Repo**[1]. Full implementation details are in Appendix B.

[1] https://anonymous.4open.science/r/ReFocus-2889

**Baseline setting** We carefully choose **TEN** well-acknowledged forecasting models as our baselines, including 1) Transformer-based methods: iTransformer (Liu et al., 2024b), Crossformer (Zhang & Yan, 2023), PatchTST (Nie et al., 2023); 2) Linear-based methods: TSMixer (Chen et al., 2023), DLinear (Zeng et al., 2023); 3) TCN-based methods: TimesNet (Wu et al., 2023), ModernTCN (Donghao & Xue, 2024); 4)Recent cutting-edge frequency-based methods that discussed earlier: FilterNet (Yi et al., 2024), FITS (Xu et al., 2024b), FreTS (Yi et al., 2023c). These models represent the latest advancements in multivariate time series forecasting and encompass all mainstream prediction model types. The results of ModernTCN, FilterNet, FITS, and FreTS are taken from FilterNet (Yi et al., 2024) and other results are taken from iTransformer (Liu et al., 2024b).

## 4.2 EXPERIMENT RESULTS

Table 1: Multivariate forecasting results with prediction lengths $F \in \{96, 192, 336, 720\}$ and fixed lookback length $T = 96$. Results are averaged from all prediction lengths. The best is **Red** and the second is Blue. The **Lower** MSE/MAE indicates the better prediction result. Full results are in Appendix K.1.

| Models | ReFocus (Ours) | | FilterNet (2024) | | iTransformer (2024b) | | ModernTCN (2024) | | FITS (2024b) | | PatchTST (2023) | | Crossformer (2023) | | TimesNet (2023) | | TSMixer (2023) | | DLinear (2023) | | FreTS (2023c) | |
|---|---|---|---|---|---|---|---|---|---|---|---|---|---|---|---|---|---|---|---|---|---|
| Metric | MSE | MAE | MSE | MAE | MSE | MAE | MSE | MAE | MSE | MAE | MSE | MAE | MSE | MAE | MSE | MAE | MSE | MAE | MSE | MAE | MSE | MAE |
| ETTm1 | **0.387** | **0.394** | 0.392 | 0.401 | 0.407 | 0.410 | 0.389 | 0.402 | 0.415 | 0.408 | 0.387 | 0.400 | 0.513 | 0.496 | 0.400 | 0.406 | 0.398 | 0.407 | 0.403 | 0.407 | 0.408 | 0.416 |
| ETTm2 | **0.275** | **0.320** | 0.285 | 0.328 | 0.288 | 0.332 | 0.279 | 0.322 | 0.286 | 0.328 | 0.281 | 0.326 | 0.757 | 0.610 | 0.291 | 0.333 | 0.289 | 0.333 | 0.350 | 0.401 | 0.321 | 0.368 |
| ETTh1 | **0.434** | **0.433** | 0.441 | 0.439 | 0.454 | 0.447 | 0.446 | 0.433 | 0.451 | 0.440 | 0.469 | 0.454 | 0.529 | 0.522 | 0.458 | 0.450 | 0.463 | 0.452 | 0.456 | 0.452 | 0.475 | 0.463 |
| ETTh2 | **0.371** | **0.396** | 0.383 | 0.407 | 0.383 | 0.407 | 0.382 | 0.404 | 0.383 | 0.408 | 0.387 | 0.407 | 0.942 | 0.684 | 0.414 | 0.427 | 0.401 | 0.417 | 0.559 | 0.515 | 0.472 | 0.465 |
| ECL | **0.168** | **0.262** | 0.173 | 0.268 | 0.178 | 0.270 | 0.197 | 0.282 | 0.217 | 0.295 | 0.205 | 0.290 | 0.244 | 0.334 | 0.192 | 0.295 | 0.186 | 0.287 | 0.212 | 0.300 | 0.189 | 0.278 |
| Traffic | **0.412** | **0.265** | 0.463 | 0.310 | 0.428 | 0.282 | 0.546 | 0.348 | 0.627 | 0.376 | 0.481 | 0.304 | 0.550 | 0.304 | 0.620 | 0.336 | 0.522 | 0.357 | 0.625 | 0.383 | 0.618 | 0.390 |
| Weather | **0.245** | **0.271** | 0.245 | 0.272 | 0.258 | 0.279 | 0.247 | 0.272 | 0.249 | 0.276 | 0.259 | 0.281 | 0.259 | 0.315 | 0.259 | 0.287 | 0.256 | 0.279 | 0.265 | 0.317 | 0.250 | 0.270 |
| Solar_Energy | **0.222** | **0.252** | 0.243 | 0.281 | 0.233 | 0.262 | 0.244 | 0.286 | 0.395 | 0.407 | 0.270 | 0.307 | 0.641 | 0.639 | 0.301 | 0.319 | 0.260 | 0.297 | 0.330 | 0.401 | 0.248 | 0.296 |

**Quantitative comparison** Comprehensive forecasting results are listed in Table 1. We leave full forecasting results in APPENDIX to save place. It is quite evident that **ReFocus** has demonstrated superior predictive performance across all datasets, significantly outperforming the second-best method. Especially, Compared to the previous SOTA **iTransformer**, we have reduced the MSE by **4%**, **6%**, and **5%** on the three most challenging benchmarks: Traffic, ECL, and Solar, respectively, indicating a significant breakthrough. These significant improvements indicate that the **ReFocus** model possesses robust performance and broad applicability in multivariate time series forecasting tasks, especially in tasks with a large number of channels, such as the Solar_Energy dataset (**137** channels), ECL dataset (**321** channels), and Traffic dataset (**862** channels).

## 4.3 MODEL ANALYSIS

Table 2: Ablation of 'Adaptive Mid-Frequency Energy Optimizer (**AMEO**)' and 'Key-Frequency Enhanced Training strategy (**KET**)'. We list the average results. Full results are in Appendix K.2.

| AMEO | KET | ETTm1 | | ETTm2 | | ETTh1 | | ETTh2 | | ECL | | Traffic | | Weather | | Solar_Energy | |
|---|---|---|---|---|---|---|---|---|---|---|---|---|---|---|---|---|---|
| | | MSE | MAE | MSE | MAE | MSE | MAE | MSE | MAE | MSE | MAE | MSE | MAE | MSE | MAE | MSE | MAE |
| - | - | 0.401 | 0.403 | 0.283 | 0.325 | 0.440 | 0.437 | 0.376 | 0.400 | 0.178 | 0.270 | 0.449 | 0.289 | 0.252 | 0.278 | 0.232 | 0.264 |
| - | ✓ | 0.394 | 0.396 | 0.279 | 0.322 | 0.437 | 0.435 | 0.373 | 0.398 | 0.171 | 0.263 | 0.414 | 0.268 | 0.250 | 0.275 | 0.228 | 0.258 |
| ✓ | - | 0.393 | 0.402 | 0.282 | 0.326 | 0.443 | 0.440 | 0.372 | 0.397 | 0.174 | 0.267 | 0.452 | 0.289 | 0.248 | 0.275 | 0.231 | 0.261 |
| ✓ | ✓ | **0.387** | **0.394** | **0.275** | **0.320** | **0.434** | **0.433** | **0.371** | **0.396** | **0.168** | **0.262** | **0.412** | **0.265** | **0.245** | **0.271** | **0.222** | **0.252** |

**Ablation study of AMEO and KET** To evaluate the contributions of each module in ReFocus, we performed ablation studies on the 'Adaptive Mid-Frequency Energy Optimizer (AMEO)' and the 'Key-Frequency Enhanced Training (KET)' strategy. The results are summarized in Table 2. Notably, integrating both modules achieves the best performance, highlighting the effectiveness of their synergy. Additionally, each module delivers substantial improvements over baseline models in most cases.

**Further study of KET** We conducted further ablation studies on the KET to demonstrate the importance of alternate training between real and synthetic data. The experimental results in

Table 3: Further ablation of 'Key-Frequency Enhanced Training strategy (**KET**)'. 'Real' means KET is not performed, i.e. trained on original data. 'Pseudo' means trained on Pseudo samples. If both are used (Bottom Line), this means the model is trained on Real and Pseudo samples alternatively, i.e. **KET**. We list the average results. Full results are in Appendix K.3.

| Real | Pseudo | ETTm1 | | ETTm2 | | ETTh1 | | ETTh2 | | ECL | | Traffic | | Weather | | Solar_Energy | |
|------|--------|-------|-----|-------|-----|-------|-----|-------|-----|-----|-----|---------|-----|---------|-----|--------------|-----|
| | | MSE | MAE | MSE | MAE | MSE | MAE | MSE | MAE | MSE | MAE | MSE | MAE | MSE | MAE | MSE | MAE |
| ✓ | - | 0.401 | 0.403 | 0.283 | 0.325 | 0.440 | 0.437 | 0.376 | 0.400 | 0.178 | 0.270 | 0.449 | 0.289 | 0.252 | 0.278 | 0.232 | 0.264 |
| - | ✓ | 0.396 | 0.398 | 0.280 | 0.323 | **0.436** | **0.434** | **0.372** | **0.397** | 0.175 | 0.266 | 0.417 | 0.271 | 0.252 | 0.276 | 0.277 | 0.294 |
| ✓ | ✓ | **0.394** | **0.396** | **0.279** | **0.322** | 0.437 | 0.435 | 0.373 | 0.398 | **0.171** | **0.263** | **0.414** | **0.268** | **0.250** | **0.275** | **0.228** | **0.258** |

Table 3 reveal that while training on pseudo samples can partially enhance the model's generalization performance on the test set, it also tends to cause over-smoothing and training instability on more complex datasets, such as Solar_Energy. In contrast, training on real and synthetic data alternatively (KET) improves generalization and mitigates over-smoothing and training instability by preserving the spectral dependencies of real samples. More Analyses are in Appendix C.

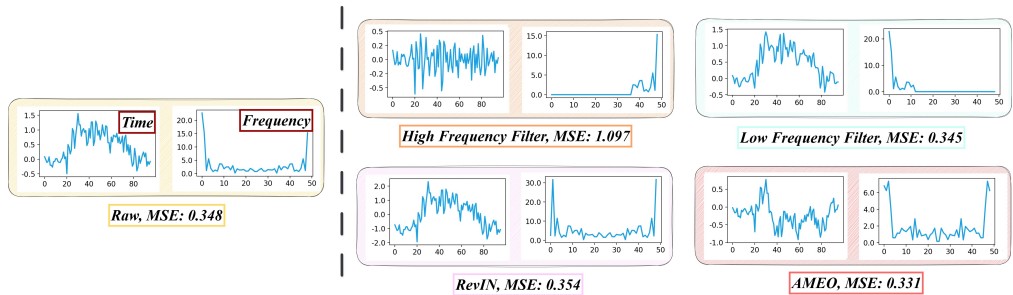

Figure 4: The time-frequency domain visualization of the original sequence (ETTm1, *the last variate*), the sequence processed by high-pass and low-pass filters, by RevIN, and by AMEO. We selected the $input - 96 - forecast - 96$ task.

**Superiority of AMEO over RevIN and Filters**     We investigated the roles of AMEO, RevIN, and Filters in addressing the Mid-Frequency Spectrum Gap through time-frequency domain visualization analysis. The results presented in Figure 4 align perfectly with our previous theoretical analysis. High-pass and low-pass filters fail to address the Mid-Frequency Spectrum Gap and exacerbate this issue. RevIN, on the other hand, merely eliminates the energy of the zero-frequency component while scaling other components using the variance $\sigma^2$, which also does not effectively resolve the problem. In contrast, our AMEO successfully amplifies the mid-frequency energy. Furthermore, compared to the original sequence and the sequence processed by RevIN, we observe that the sequence processed by AMEO exhibits significantly higher stationarity with much more stable means and variance. More Analyses are in Appendix F.

**Outstanding inter-series modeling ability of the EKPB**     In the multivariate correlation analysis in Figure 5 (**LEFT**), the early encoder layer produces correlation maps similar to the input series $X$. In deeper layers, these maps gradually resemble the correlation patterns of the target series $Y$, suggesting that ReFocus effectively models inter-series dependencies in a hierarchical and progressive manner. Furthermore, Figure 5 (**RIGHT**) indicates that ReFocus effectively captures Key-Frequency shared across channels. To illustrate EKPB's functionality, we visualize the series embeddings with and without its adjustment in Figure 6 (**LEFT**). The T-SNE visualization of the series embeddings shows that without EKPB, using only the channel-independent strategy (Nie et al., 2023), the MSE is 0.171. After applying EKPB, channels sharing Key-Frequency (variates 2&3) are clustered, while others (variates 1&3) are separated. This adjustment improves the MSE from 0.171 to 0.145, a **15%** reduction.

**Efficiency Analysis of ReFocus**

ReFocus delivers higher performance with minimal memory and time consumption. We left the detailed complexity analysis and more information in Appendix H, which shows that our ReFcous with only **Linear** complexity. It achieves better performance with significantly lower resource

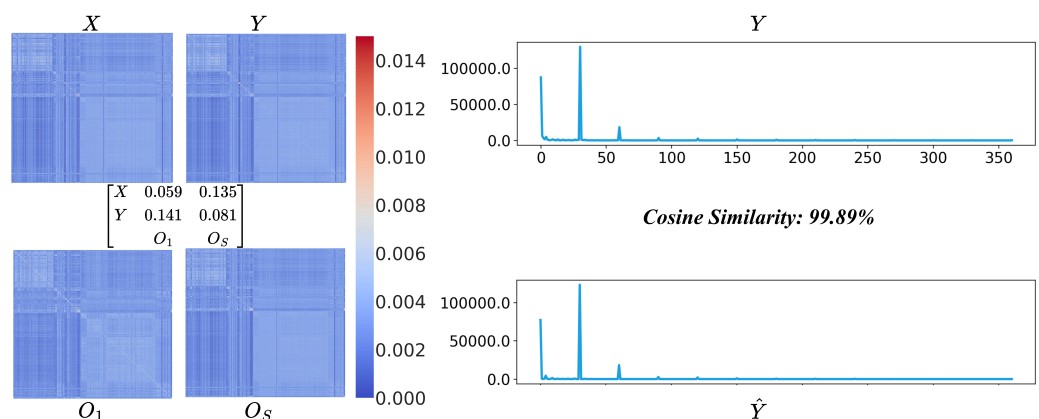

Figure 5: Analysis of multivariate correlation (Left) and Shared Key-Frequency (Right) on ECL, with *input-96-forecast-720*. **Left**: Visualization of multivariate correlations of raw time series $(X, Y)$ and the learned embedding $(O_1, O_S)$. $O_1$ is the output embedding of the first encoder block, and $O_S$ the last block. Frobenius norm $\|A - B\|_F = \sqrt{\sum_{i,j}(A_{ij} - B_{ij})^2}$ is used to quantify the similarity (shown in Figure center). The lower, the higher. More examples and details are in Appendix G. **Right**: We visualize the average spectral energy across all channels for both the ground truth samples $Y$ and the predictions $\hat{Y}$.

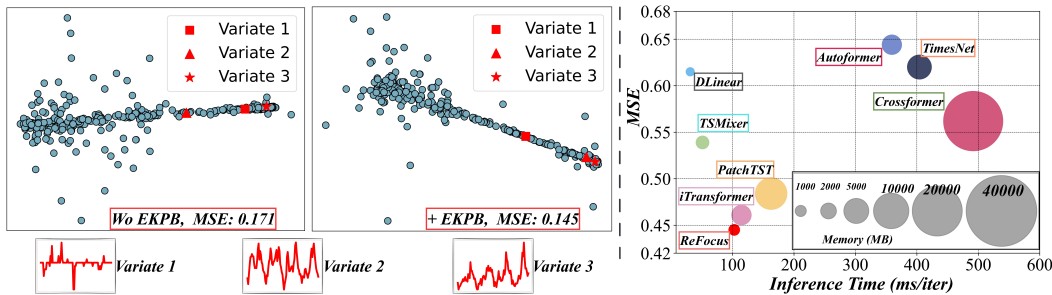

Figure 6: **Left**: T-SNE visualization of the series embeddings with and without 'Energy-based Key-Frequency Picking Block' (EKPB) on ECL. We choose the $input - 96 - forecast - 96$ task. Three example variates are highlighted: variates 2&3 shared a common Key-Frequency, while variate 1 does not. **Right**: Memory and time consumption of different models. CoIn-based models demonstrate efficient performance with minimal memory and computational overhead.

consumption, as in Figure 6 (**RIGHT**). ReFocus remains competitive even against other Linear-based or MLP-based approaches, such as DLinear and TSMixer.

## 5 CONCLUSION

This work addresses two critical challenges in multivariate time series forecasting: the Mid-Frequency Spectrum Gap and the efficient modeling of the shared Key-Frequency. We propose the 'Adaptive Mid-Frequency Energy Optimizer', which effectively enhances mid-frequency extraction, and the 'Energy-based Key-Frequency Picking Block' with the 'Key-Frequency Enhanced Training' strategy, which efficiently captures shared frequency patterns. Extensive experiments demonstrate the superiority of our approach, achieving up to 6% MSE reduction on challenging benchmarks, thus advancing the SOTA in frequency-domain forecasting.

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

# A    PROOF

This section is dedicated to proving Theorem 3.3 and Theorem 3.5.

## A.1    IMPACT OF REVIN ON FREQUENCY SPECTRUM

RevIN (Kim et al., 2022; Liu et al., 2022c) normalizes inputs using sample-wise mean and variance, then reverts scaling post-prediction to ensure consistent distributions, mitigating non-stationary effects in time series.

Let the original time series be $x(t)$ with length $T$. The series $\hat{x}(t)$ that processed by RevIN is given by:

$$\hat{x}(t) = \frac{x(t) - \mu}{\sigma}, t = 0, 1, \ldots, T-1,$$

$$\mu = \frac{1}{T} \sum_{t=0}^{T-1} x(t), \quad \sigma = \sqrt{\frac{1}{T} \sum_{t=0}^{T-1} (x(t) - \mu)^2}. \tag{8}$$

The Fourier transform of $x(t)$ and $\hat{x}(t)$ are:

$$X(f) = \sum_{t=0}^{T-1} x(t) e^{-i2\pi ft/T-1}, \quad f = 0, 1, \ldots, T-1,$$

$$\hat{X}(f) = \sum_{t=0}^{T-1} \left( \frac{x(t) - \mu}{\sigma} \right) e^{-i2\pi ft/T-1}$$

$$= \frac{1}{\sigma} \sum_{t=0}^{T-1} x(t) e^{-i2\pi ft/T-1} - \frac{\mu}{\sigma} \sum_{t=0}^{T-1} e^{-i2\pi ft/T-1}. \tag{9}$$

The spectral energy is computed as the squared magnitude of the Fourier transform. For $x(t)$ and $\hat{x}(t)$, we have:

$$E_X(f) = |X(f)|^2, \quad E_{\hat{X}}(f) = |\hat{X}(f)|^2. \tag{10}$$

When $f = 0$, the exponential term $e^{-i2\pi ft/T-1} = 1$, so:

$$\hat{X}(0) = \frac{1}{\sigma} \sum_{t=0}^{T-1} x(t) - \frac{\mu T}{\sigma}$$

$$= \frac{\mu T}{\sigma} - \frac{\mu T}{\sigma}$$

$$= 0 \tag{11}$$

Since $\frac{\mu}{\sigma}$ is a constant, we have:

$$\frac{\mu}{\sigma} \cdot \sum_{t=0}^{T-1} e^{-i2\pi ft/T-1} = 0, \quad f = 1, 2 \ldots, T-1,$$

$$\hat{X}(f) = \frac{1}{\sigma} \sum_{t=0}^{T-1} x(t) e^{-i2\pi ft/T-1} - \frac{\mu}{\sigma} \sum_{t=0}^{T-1} e^{-i2\pi ft/T-1}$$

$$= \frac{1}{\sigma} X(f),$$

$$E_{\hat{X}}(f) = \left( \frac{1}{\sigma} \right)^2 |X(f)|^2. \tag{12}$$

**This suggests that RevIN scales the spectral energy by $\sigma^2$ but does not affect its relative distribution except $\hat{X}(0) = 0$.** Thus, RevIN preserves the relative spectral energy distribution and leaves the Mid-Frequency Spectrum Gap unresolved.

## A.2 IMPACT OF AMEO ON FREQUENCY SPECTRUM

Referring back to Definition 3.4, AMEO is defined as:

$$\hat{x}(t) = x(t) - \frac{\beta}{K} \sum_{k=0}^{K-1} \tilde{x}(t + K - 1 - k),$$

$$\tilde{x}(t) = \begin{cases} x(t - (\frac{K}{2} + 1)), & \text{if } \frac{K}{2} + 1 \le t < T + \frac{K}{2} + 1 \\ 0, & \text{if } 0 \le t < \frac{K}{2} + 1 \text{ or } T + \frac{K}{2} + 1 \le t < T + K \end{cases} \tag{13}$$

The Fourier transform of $\hat{x}(t)$ is:

$$\hat{X}(f) = \sum_{t=0}^{T-1} \left[ x(t) - \frac{\beta}{K} \sum_{k=0}^{K-1} \tilde{x}(t + K - 1 - k) \right] e^{-i2\pi ft/T-1}$$

$$= \underbrace{\sum_{t=0}^{T-1} x(t) e^{-i2\pi ft/T-1}}_{X(f)} - \frac{\beta}{K} \sum_{k=0}^{K-1} \underbrace{\sum_{t=0}^{T-1} \tilde{x}(t + K - 1 - k) e^{-i2\pi ft/T-1}}_{T_k(f)}. \tag{14}$$

For $T_k(f)$, given $FFT\{x(t-a)\} = X(f)e^{-i2\pi fa/T-1}$, we have:

$$T_k(f) = \sum_{t=0}^{T-1} \tilde{x}(t + K - 1 - k) e^{-i2\pi ft/T-1}$$

$$= \sum_{t=0}^{T-1} x(t + \frac{K}{2} - k - 2) e^{-i2\pi ft/T-1}$$

$$= FFT\{x(t + \frac{K}{2} - k - 2)\}$$

$$= X(f)e^{i2\pi f(\frac{K}{2} - k - 2)/T-1} \tag{15}$$

So, we have the Fourier transform of $\hat{x}(t)$ and its spectral energy:

$$\hat{X}(f) = X(f) - \frac{\beta}{K} \sum_{k=0}^{K-1} X(f) e^{i2\pi f(\frac{K}{2} - k - 2)/T-1}$$

$$= X(f) \left[ 1 - \beta \cdot \underbrace{\frac{1}{K} \sum_{k=0}^{K-1} e^{i2\pi f(\frac{K}{2} - k - 2)/T-1}}_{G(f)} \right],$$

$$E_{\hat{X}}(f) = |X(f)|^2 \left\{ 1 - \beta \cdot \underbrace{\frac{1}{K} \sum_{k=0}^{K-1} e^{i2\pi f(\frac{K}{2} - k - 2)/T-1}}_{G(f)} \right\}^2$$

$$= |X(f)|^2 (1 - \beta \cdot G(f))^2. \tag{16}$$

In this paper, we set $K = 25$ (i.e., $T/4 + 1$, $T = 96$), and the function graph of $G(f)$ is shown in Figure 7.

It is evident that $G(f)$ is a gradually decay function, with its values decreasing **from 1 to 0**. This ensures that $E_{\hat{X}}(f) = |X(f)|^2(1 - \beta \cdot G(f))^2$, where, relative to $E_X$, the low-frequency components are attenuated, and the mid-frequency components are enhanced.

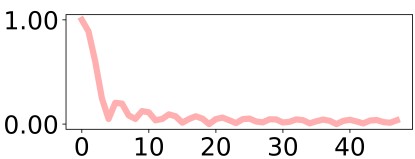

Figure 7: The function $G(f)$ is plotted for $T = 96$ and $K = 25$. Due to the symmetry of the $FFT$, we only need to plot the values for $f = 0, 1, \ldots, 48$.

# B EXPERIMENTAL DETAILS

## B.1 DATASET STATISTICS

We elaborate on the datasets employed in this study with the following details.

- **ETT Dataset** (Zhou et al., 2022a) comprises two sub-datasets: **ETTh** and **ETTm**, which were collected from electricity transformers. Data were recorded at 15-minute and 1-hour intervals for ETTm and ETTh, respectively, spanning from July 2016 to July 2018.
- **Solar_Energy** (Lai et al., 2018) records the solar power production of 137 PV plants in 2006, which are sampled every 10 minutes.
- **Electricity Dataset**[2] encompasses the electricity consumption data of 321 customers, recorded on an hourly basis, covering the period from 2012 to 2014.
- **Traffic Dataset**[3] consists of hourly data from the California Department of Transportation. It describes road occupancy rates measured by various sensors on San Francisco Bay area freeways.
- **Weather Dataset**[4] contains records of 21 meteorological indicators, updated every 10 minutes throughout the entire year of 2020.

We follow the same data processing and train-validation-test set split protocol used in iTransformer (Liu et al., 2024b), where the train, validation, and test datasets are strictly divided according to chronological order to make sure there are no data leakage issues. We fix the input length as $T = 96$ for all datasets, and the forecasting length $F \in \{96, 192, 336, 720\}$. The characteristics of these datasets are shown in Table 4

Table 4: The Statistics of the eight datasets used in our experiments.

| Datasets | ETTh1&2 | ETTm1&2 | Traffic | Electricity | Solar_Energy | Weather |
|---|---|---|---|---|---|---|
| Variates | 7 | 7 | 862 | 321 | 137 | 21 |
| Timesteps | 17,420 | 69,680 | 17,544 | 26,304 | 52,560 | 52,696 |
| Granularity | 1 hour | 5 min | 1 hour | 1 hour | 10 min | 10 min |

## B.2 IMPLEMENTATION DETAILS AND MODEL PARAMETERS

We trained our ReFocus model using the MSE loss function and employed the ADAM optimizer. For evaluation purposes, we used two key performance metrics: the mean square error (MSE) and the mean absolute error (MAE). We initialized the random seed as $rs = 2024$ and set the hyperparameter $K = 25$-kernel size of the convolution kernel in AMEO. The dimension of the Layer is set to $D = 512$ and $Q = 128$. The batch size $bs = 32$ for the Traffic dataset due to its large channel will cause **out of memory** when employed with large batch size, and $bs = 128$ for others. The learning rate is searched from $lr \in \{1e-5, 1e-4\}$ except for the Traffic dataset ($lr = 5e-4$). The number of EKPB is searched from $N \in \{1, 2, 3, 4\}$, and hyperparameter $\beta$. which controls the scale magnitude, from $\beta \in \{0.01, 0.1, 0.5, 1.0\}$. Our implementation was carried out in PyTorch and

---

[2]https://archive.ics.uci.edu/ml/datasets/ElectricityLoadDiagrams20112014
[3]https://pems.dot.ca.gov/
[4]https://www.bgc-jena.mpg.de/wetter/

executed on a single NVIDIA GeForce RTX 4090 with 24G VRAM. To foster reproducibility, we make our code, and training scripts available in this **Anonymous Repo**[5].

All the compared multivariate forecasting baseline models that we reproduced are implemented based on the benchmark of **Time series Lab** (Wang et al., 2024d) Repository [6], which is fairly built on the configurations provided by each model's original paper or official code. Those that have not yet been included in **Time series Lab** are directly reproduced from their official code repositories. It is worth noting that both the baselines used in this paper and our **ReFocus** have fixed a long-standing bug. This bug was originally identified in Informer (Zhou et al., 2022a) **(AAAI 2021 Best Paper)** and subsequently addressed by FITS (Xu et al., 2024b). For specific details about the bug and its resolution, please refer to **GitHub Repository**[7].

## C  Further Analysis of the proposed Key-Frequency Enhanced Training strategy

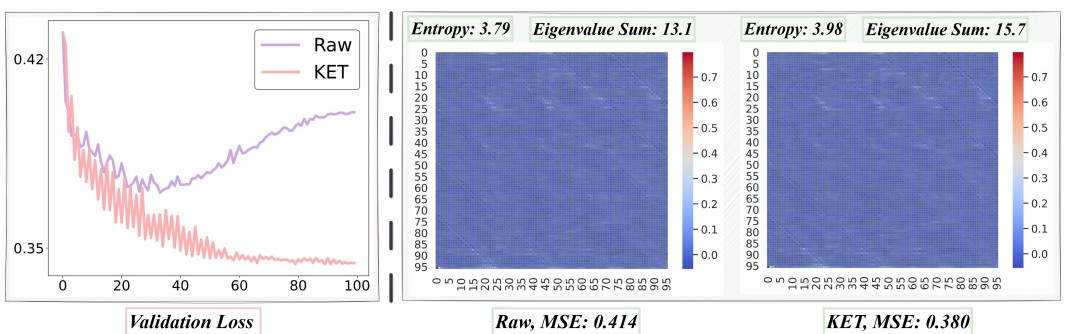

Figure 8: We select the $input - 96 - forecast - 96$ task on Traffic and visualize the validation loss and weight of our **ReFocus** model. **LEFT**: Visualization of the **Validation Loss** during 100 training epochs with (**KET**) and without KET (**Raw**). **RIGHT**: Visualization about the Weight (Obtained using the approach outlined in **Analysis of linear model** (Toner & Darlow, 2024)) of the trained model. Two significant metrics for assessing the information richness of the weight matrix-the information **Entropy** and the **Sum of Eigenvalues**-are calculated. Both indicate higher quality with greater values.

To further investigate the impact of the proposed **'Key-Frequency Enhanced Training (KET) strategy'** on model training and forecasting ability, we visualize its training process regarding Validation Loss and the model weights obtained after training in Figure 8. We also compute the **Entropy** and the **Sum of Eigenvalues** of the weight matrix.

The results show that, in the absence of KET, the model quickly overfits around the **24**th epoch, exhibiting poor generalization. In contrast, with the aid of KET, the model consistently performs better on the validation set, converging smoothly without overfitting, and the training process becomes more stable. Additionally, weight visualization results indicate that the model trained with KET has higher information **Entropy** and a greater **Sum of Eigenvalues**, suggesting that the trained model possesses a stronger capacity for feature representation extraction. The predictive results further validate this, as our KET improves the MSE from 0.414 to 0.380, achieving an **8.2%** reduction.

## D  Ablation Study of Different Key-Frequency Picking Strategy

We conducted an ablation study on various key-frequency selection strategies. The evaluated methods include Maximum-based, Minimum-based, and **Softmax-based** random sampling strategies. Our experimental results in Table 5 reveal that purely relying on Maximum or Minimum-based strategies may overlook certain critical Key-Frequency. In contrast, the random sampling strategy based on a Softmax probabilistic distribution consistently achieved the best overall performance, particularly on

---

[5]https://anonymous.4open.science/r/ReFocus-2889/
[6]https://github.com/thuml/Time-Series-Library
[7]https://github.com/VEWOXIC/FITS

Table 5: Ablation study of different Key-Frequency Picking strategies. 'Softmax' means using softmax function to generate a probability distribution and picking shared Key-Frequency using this distribution. 'Max' means always choosing the biggest energy. 'Min' means always choosing the smallest energy. We list the average results. Full results are in Appendix K.4.

| Picking Strategy | ETTm1 | | ETTm2 | | ETTh1 | | ETTh2 | | ECL | | Traffic | | Weather | | Solar_Energy | |
|---|---|---|---|---|---|---|---|---|---|---|---|---|---|---|---|---|
| | MSE | MAE | MSE | MAE | MSE | MAE | MSE | MAE | MSE | MAE | MSE | MAE | MSE | MAE | MSE | MAE |
| Min | 0.388 | **0.392** | 0.280 | 0.323 | **0.432** | **0.432** | 0.371 | 0.396 | 0.194 | 0.281 | 0.517 | 0.344 | 0.378 | 0.363 | 0.240 | 0.270 |
| Max | 0.392 | 0.395 | 0.279 | 0.322 | 0.437 | 0.435 | 0.374 | 0.398 | 0.172 | 0.265 | 0.422 | 0.273 | 0.351 | 0.343 | 0.230 | 0.260 |
| **Softmax** | **0.387** | 0.394 | **0.275** | **0.320** | 0.434 | 0.433 | **0.371** | **0.396** | **0.168** | **0.262** | **0.412** | **0.265** | **0.346** | **0.339** | **0.222** | **0.252** |

datasets with a larger number of channels and higher complexity—key challenges in multivariate time series forecasting.

# E COMPARISON OF EKPB AND OTHER INTER-SERIES DEPENDENCIES MODELING BACKBONE

Table 6: Multivariate forecasting result of 'Energy-based Key-Frequency Picking Block' (EKPB) and other inter-series dependencies modeling backbones. We use prediction lengths $F \in \{96, 192, 336, 720\}$, and input length $T = 96$. The best results are in **bold**.

| Model | | **EKPB** | | iTransformer | | TSMixer | | Crossformer | | FECAM | |
|---|---|---|---|---|---|---|---|---|---|---|---|
| Metric | | MSE | MAE | MSE | MAE | MSE | MAE | MSE | MAE | MSE | MAE |
| ETTm2 | 96 | **0.179** | **0.260** | 0.180 | 0.264 | 0.182 | 0.266 | 0.287 | 0.366 | 0.188 | 0.275 |
| | 192 | **0.244** | **0.301** | 0.250 | 0.309 | 0.249 | 0.309 | 0.414 | 0.492 | 0.265 | 0.336 |
| | 336 | **0.303** | **0.339** | 0.311 | 0.348 | 0.309 | 0.347 | 0.597 | 0.542 | 0.318 | 0.362 |
| | 720 | **0.401** | **0.395** | 0.412 | 0.407 | 0.416 | 0.408 | 1.730 | 1.042 | 0.416 | 0.417 |
| | Avg | **0.282** | **0.324** | 0.288 | 0.332 | 0.289 | 0.333 | 0.757 | 0.610 | 0.297 | 0.348 |
| ETTh2 | 96 | **0.288** | **0.338** | 0.297 | 0.349 | 0.319 | 0.361 | 0.745 | 0.584 | 0.298 | 0.345 |
| | 192 | **0.374** | **0.391** | 0.380 | 0.400 | 0.402 | 0.410 | 0.877 | 0.656 | 0.377 | 0.397 |
| | 336 | **0.414** | **0.426** | 0.428 | 0.432 | 0.444 | 0.446 | 1.043 | 0.731 | 0.425 | 0.434 |
| | 720 | **0.421** | **0.440** | 0.427 | 0.445 | 0.441 | 0.450 | 1.104 | 0.763 | 0.432 | 0.450 |
| | Avg | **0.374** | **0.399** | 0.383 | 0.407 | 0.401 | 0.417 | 0.942 | 0.684 | 0.383 | 0.407 |
| Weather | 96 | 0.166 | **0.209** | 0.174 | 0.214 | 0.166 | 0.210 | **0.158** | 0.230 | 0.182 | 0.242 |
| | 192 | 0.216 | 0.256 | 0.221 | **0.254** | 0.215 | 0.256 | **0.206** | 0.277 | 0.223 | 0.281 |
| | 336 | 0.274 | **0.296** | 0.278 | 0.296 | 0.287 | 0.300 | 0.272 | 0.335 | **0.270** | 0.320 |
| | 720 | 0.351 | **0.346** | 0.358 | 0.349 | 0.355 | 0.348 | 0.398 | 0.418 | **0.338** | 0.374 |
| | Avg | **0.252** | **0.277** | 0.258 | 0.279 | 0.256 | 0.279 | 0.259 | 0.315 | 0.253 | 0.304 |
| ECL | 96 | **0.146** | **0.240** | 0.148 | 0.240 | 0.157 | 0.260 | 0.219 | 0.314 | 0.178 | 0.267 |
| | 192 | **0.161** | 0.254 | 0.162 | **0.253** | 0.173 | 0.274 | 0.231 | 0.322 | 0.185 | 0.273 |
| | 336 | **0.178** | 0.273 | 0.178 | **0.269** | 0.192 | 0.295 | 0.246 | 0.337 | 0.199 | 0.290 |
| | 720 | **0.220** | **0.306** | 0.225 | 0.317 | 0.223 | 0.318 | 0.280 | 0.363 | 0.235 | 0.323 |
| | Avg | **0.176** | **0.268** | 0.178 | 0.270 | 0.186 | 0.287 | 0.244 | 0.334 | 0.199 | 0.288 |

Table 6 presents the full results of 'Energy-based Key-Frequency Picking Block (EKPB)' and other inter-series dependency modeling backbones on multivariate time series forecasting tasks. We compared 'Energy-based Key-Frequency Picking Block' (EKPB) with several well-established backbones, including iTransformer (Liu et al., 2024b), TSMixer (Chen et al., 2023), and Crossformer (Zhang & Yan, 2023), which have demonstrated exceptional performance in modeling inter-series dependencies. Additionally, we included FECAM (Jiang et al., 2023), a method also designed for modeling cross-channel frequency-domain dependencies. The results presented in Table 6 demonstrate that our EKPB outperforms in modeling inter-series dependencies across multiple datasets.

# F ABLATION STUDY OF DIFFERENT FREQUENCY PROCESSING STRATEGY

In Table 7, the performance of AMEO on two prediction tasks across two datasets consistently surpasses the results achieved by methods based on RevIN and Filters. Furthermore, while Filters and

Table 7: Experiment result of high-pass filter, low-pass filter, RevIN, and AMEO using **a simple linear projection** as the forecaster on Weather and ETTm1 dataset. We set the input length $T = 96$ and forecasting length $F \in \{720, 96\}$.

| Dataset | Length | AMEO | | RevIN | | Low | | High | | None | |
|---|---|---|---|---|---|---|---|---|---|---|---|
| | | MSE | MAE | MSE | MAE | MSE | MAE | MSE | MAE | MSE | MAE |
| ETTm1 | 96 | **0.331** | **0.365** | 0.354 | 0.375 | 0.345 | 0.371 | 1.097 | 0.792 | 0.348 | 0.375 |
| | 720 | **0.466** | **0.440** | 0.486 | 0.448 | 0.478 | 0.458 | 1.106 | 0.796 | 0.479 | 0.456 |
| Weather | 96 | **0.164** | **0.236** | 0.194 | 0.234 | 0.198 | 0.258 | 0.636 | 0.608 | 0.198 | 0.258 |
| | 720 | **0.331** | **0.370** | 0.365 | 0.353 | 0.353 | 0.387 | 0.638 | 0.611 | 0.352 | 0.386 |

RevIN occasionally lead to degraded performance on certain datasets, AMEO consistently delivers results that outperform the original methods. These findings further highlight the superiority of AMEO over alternative approaches.

## G  VISUALIZATION OF MULTIVARIATE CORRELATIONS

In addition to the main results, we further provide visualizations of multivariate correlations on the Traffic datasets, as shown in Figure 9 (**LEFT**). We calculate the value of the correlation map by:

$$\bar{x}_{i,:} = \frac{1}{T} \sum_{t=1}^{T} x_{i,t}, \quad \tilde{x}_{i,t} = x_{i,t} - \bar{x}_{i,:},$$

$$\tilde{\rho}_{ij} = \frac{\sum_{t=1}^{T} \tilde{x}_{i,t}\, \tilde{x}_{j,t}}{\sqrt{\sum_{t=1}^{T} \tilde{x}_{i,t}^2}\, \sqrt{\sum_{t=1}^{T} \tilde{x}_{j,t}^2} + \epsilon}, \quad \epsilon = 1 \times 10^{-8},$$

$$\text{cor}_{ij} = \frac{\exp\left(\tilde{\rho}_{ij}\right)}{\sum_{k=1}^{N} \exp\left(\tilde{\rho}_{ik}\right)}. \tag{17}$$

These results demonstrate that ReFocus effectively captures inter-series dependencies. Moreover, the encoder layer built based on the 'Energy-based Frequency Picking Block' (EKPB) exhibits highly interpretable feature maps. Notably, in the shallow layers, the correlation maps of the learned representation closely resemble the input sequence $X$, while in deeper layers, the representations progressively align with the target outputs $Y$. This suggests that ReFocus progressively transforms input features toward task-relevant representations, enabling both effective modeling and interpretability.

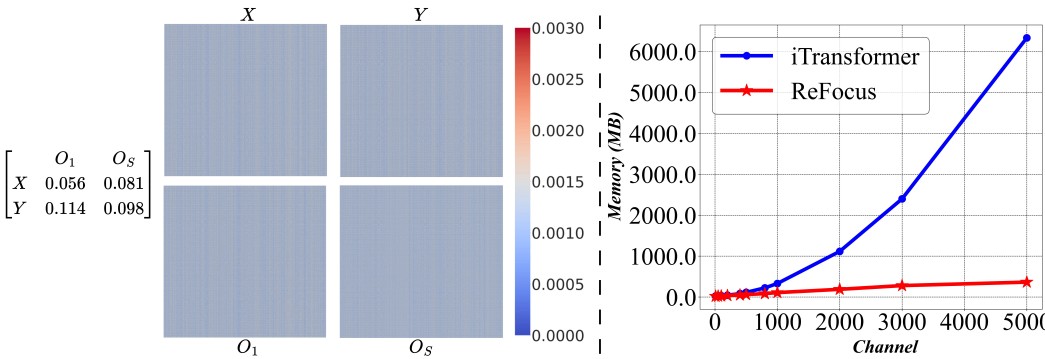

Figure 9: **Left**: Visualization of multivariate correlations of raw time series $(X, Y)$ and the learned embedding. $O_1$ is the output embedding of the first encoder block, and $O_S$ the last. We use the Frobenius norm $\|A - B\|_F = \sqrt{\sum_{i,j}(A_{ij} - B_{ij})^2}$ to quantify the similarity. The lower, the higher. **Right**: Memory Usage comparison of ReFocus and iTransformer. We compare the memory consumption under varying numbers of channels (with $B = 4, T = 96, F = 720$).

# H   COMPLEXITY ANALYSIS OF REFOCUS

We analyze the complexity of the ReFocus Encoder concerning the Dimension of model $D$, number of channels $N$, dimension of 'Energy-based Frequency Picking Block' (EKPB) $Q$. The complexity of input **MLP** is $O(N \cdot D^2) + O(N \cdot D \cdot Q)$. Since $Q \ll D$ ($Q = 128, D = 512$ in this work), the dominant cost is $O(N \cdot D^2)$. During **EKPB**, the complexity of *FFT* is $O(N \cdot Q \cdot \log Q)$, and so is the *IFFT*. Complexity of *Energy computing*, *Softmax* and *Component Picking*, each part is $O(N \cdot Q)$. Map the representation back to the original space: $O(N \cdot D \cdot Q)$. So the overall complexity is $O(N \cdot D \cdot Q)$, which is negligible compared to $O(N \cdot D^2)$. Each later **FFN** has $O(N \cdot D^2)$ complexity. So, the overall complexity of ReFocus Encoder is dominated by $O(N \cdot D^2)$, which is Linear to input token number $N$.

On the other hand, a standard Transformer encoder has complexity $O(N^2 \cdot D) + O(N \cdot D^2)$ (Self-Attention + FFN). For large $N$, the Quadradic $O(N^2)$ term becomes the bottleneck, whereas our **ReFocus** Block avoids this quadratic cost.

As shown in Figure 9 (**RIGHT**), we compare the memory consumption of ReFocus and iTransformer under varying numbers of channels. **ReFocus** exhibits high efficiency, achieving SOTA performance with significantly reduced memory usage and computational cost.

Table 8: Model efficiency analysis. We evaluated the **parameter count**, and the **inference time** (average of 5 runs on a single NVIDIA 4090 24GB GPU) with $batch\_size = 1$ on **ECL** dataset. We set the dimension of layer $dim \in \{256, 512\}$, and the number of network layers $N = 2$. The task is **input-96-forecast-720**. **\*** means 'former.' **Para** means 'Parameter count(M).' **Time** means 'inference time(ms).'

| Dim | EKPB | | Cross* | | iTrans* | | TSMixer | | FECAM | |
|---|---|---|---|---|---|---|---|---|---|---|
| | Param | Time | Para | Time | Para | Time | Para | Time | Para | Time |
| 256 | **0.29** | **68.91** | 0.93 | 98.37 | 1.27 | 192.12 | 13.66 | 432.40 | 1.39 | 205.66 |
| 512 | **0.97** | **84.54** | 1.78 | 118.29 | 4.63 | 249.60 | 43.04 | 507.54 | 5.14 | 277.43 |

Additionally, when comparing the number of parameters and inference time during prediction under identical configurations on the ECL dataset, our EKPB method still outperforms other inter-series dependencies modeling baselines by a significant margin, as in Table 8.

# I   SENSITIVITY OF HYPERPARAMETERS

Table 9: Sensitivity to mixing coefficient in KET ($\alpha$). Since we don't know in advance which two channels share the same frequency information and their correlation intensity, in our work, we choose to use $\alpha$ sampled from a Normal Distribution to simulate such correlation. Here, we design three ablation methods: w/o means $\alpha = 0$, Normal is our default design, and Constant denotes $\alpha = 1$.

| Dataset | w/o | Normal | Constant |
|---|---|---|---|
| ETTm2 | 0.402/0.396 | **0.395/0.392** | 0.399/0.394 |
| Weather | 0.350/0.346 | **0.344/0.343** | 0.349/0.346 |

Table 10: Sensitivity to augmentation intensity ($\beta$) in AMEO.

| Dataset | 0.0 | 0.2 | 0.4 | 0.6 | 0.8 | 1.0 |
|---|---|---|---|---|---|---|
| ETTm2 | 0.398 | 0.397 | 0.397 | 0.397 | 0.397 | **0.396** |
| Weather | 0.349 | 0.348 | **0.346** | **0.346** | 0.347 | 0.348 |

We studied the sensitivity of **ReFocus** to four major hyperparameters: the mixing coefficient in KET ($\alpha$, Table 9), the augmentation intensity parameter ($\beta$, Table 10), the kernel size ($K$, Table 11) in AMEO, and the number of encoder layers ($N$, Table 12). Results of MSE on ETTm2 and Weather datasets are reported.

Table 11: Sensitivity to kernel size ($K$) in AMEO.

| Dataset | 7 | 15 | 25 | 51 | 75 |
|---------|-----|-----|-----|-----|-----|
| ETTm2 | 0.398 | 0.399 | **0.396** | 0.397 | 0.397 |
| Weather | **0.346** | **0.346** | **0.346** | 0.349 | 0.349 |

Table 12: Sensitivity to the number of encoder layers ($N$).

| Dataset | 1 | 2 | 3 | 4 | 5 |
|---------|-----|-----|-----|-----|-----|
| ETTm2 | **0.396** | 0.397 | 0.397 | 0.398 | 0.398 |
| Weather | 0.347 | 0.346 | 0.348 | **0.344** | 0.348 |

## J   EXTENDED DISCUSSION AND EVALUATION

### J.1   BROADER DISCUSSION ON THE MID-FREQUENCY GAP

Although we only demonstrated the mid-frequency gap on energy, traffic, and weather benchmarks, this phenomenon is well-documented in many other domains. In econometrics, financial time series exhibit spectra dominated by very low frequencies with a mid-band "valley" that simple filters cannot remedy (Stock & Watson, 2002; Granger & Newbold, 1974). In biomedical signals such as EEG, negligible mid-frequency energy in resting-state recordings is commonly reported (Chatfield & Xing, 2019; Niedermeyer & da Silva, 2005). Similar patterns were also observed in human activity recognition (HAR) signals, where energy is concentrated in low-frequency bands while the mid-frequency region remains nearly empty (Anguita et al., 2013; Lara & Labrador, 2012).

To validate the broader applicability of ReFocus, we extended experiments to classification tasks in Medical time series and HAR. Following Medformer (Wang et al., 2024c), we evaluate on APAVA (EEG dataset) and UCI-HAR benchmarks, both known to exhibit mid-frequency gaps. Baselines include Medformer (Wang et al., 2024c), iTransformer (Liu et al., 2024b), and PatchTST (Nie et al., 2023), using Accuracy and F1-score as metrics. Results in Table 13 confirm the effectiveness of ReFocus across these tasks.

Table 13: Classification results on APAVA and UCI-HAR datasets.

| Dataset | Metric | ReFocus (Ours) | Medformer | iTransformer | PatchTST |
|---------|--------|----------------|-----------|--------------|----------|
| APAVA | Accuracy | **82.61** | 78.74 | 74.55 | 67.03 |
| | F1-score | **81.88** | 76.31 | 72.30 | 55.97 |
| UCI-HAR | Accuracy | **93.17** | 91.65 | 92.41 | 87.67 |
| | F1-score | **93.27** | 91.61 | 92.39 | 88.02 |

### J.2   SHORT-TERM TIME SERIES FORECASTING

We conducted experiments on the M4 benchmark to test the generalizability of ReFocus on short-term time series forecasting. Following the protocol of TimesNet (Wu et al., 2023), we compare against TimesNet (Wu et al., 2023), PatchTST (Nie et al., 2023), DLinear (Zeng et al., 2023), and Autoformer (Wu et al., 2021). Performance is reported with SMAPE, MASE, and OWA (lower is better). As shown in Table 14, ReFocus consistently achieves the best overall results across yearly, quarterly, and monthly tasks, further affirming its robustness in modeling complex temporal variations.

These extended evaluations demonstrate the broad applicability and robustness of our approach beyond the original long-term time series forecasting tasks.

Table 14: Forecasting results on the M4 dataset. Lower SMAPE/MASE/OWA is better.

| Category | Metric | ReFocus (Ours) | TimesNet | PatchTST | DLinear | Autoformer |
|---|---|---|---|---|---|---|
| Yearly | SMAPE | **13.201** | 13.387 | 16.463 | 16.965 | 13.974 |
| | MASE | **2.912** | 2.996 | 3.967 | 4.283 | 3.134 |
| | OWA | **0.778** | 0.786 | 1.003 | 1.058 | 0.822 |
| Quarterly | SMAPE | **9.964** | 10.100 | 10.644 | 12.145 | 11.338 |
| | MASE | **1.162** | 1.182 | 1.278 | 1.520 | 1.365 |
| | OWA | **0.873** | 0.890 | 0.949 | 1.106 | 1.012 |
| Monthly | SMAPE | **12.541** | 12.670 | 13.399 | 13.514 | 13.958 |
| | MASE | **0.914** | 0.933 | 1.031 | 1.037 | 1.103 |
| | OWA | **0.862** | 0.878 | 0.949 | 0.956 | 1.002 |

## K  FULL RESULTS

The full experiment results are provided in the following section due to the space limitation of the main text.

### K.1  FULL MULTIVARIATE FORECASTING RESULTS

Table 15 contains the detailed results of Ten baselines and our ReFocus on eight well-acknowledged forecasting benchmarks. ReFocus consistently achieves the best overall performance across all datasets, especially in tasks with a large number of channels, such as the Solar_Energy dataset (**137** channels), ECL dataset (**321** channels), and Traffic dataset (**862** channels). It obtains the best performance in terms of MSE: **34 out of 40** tasks, and MAE: **36 out of 40** tasks. These results demonstrate the outstanding performance of ReFocus in multivariate time series forecasting tasks.

### K.2  FULL RESULTS OF ABLATION ON AMEO AND KET

Table 16 presents the full results of the ablation study on 'Adaptive Mid-Frequency Energy Optimizer (AMEO)' and 'Key-Frequency Enhanced Training (KET)'. KET and AMEO contribute significantly to the model's performance, each providing substantial improvements. Moreover, their combination further enhances the model, achieving peak performance. These results provide strong evidence of the effectiveness of both AMEO and KET.

### K.3  FULL RESULTS OF FURTHER ABLATION STUDY ON KET

Table 17 exhibits the full results of a further ablation study on the 'Key-Frequency Enhanced Training (KET)' strategy. Introducing Pseudo samples—obtained by randomly incorporating spectral information from other channels into the current channel—generally leads to performance improvement. However, on more complex datasets, it results in performance degradation. In contrast, alternating training between Real and Pseudo samples (**Our KET**) overcomes this issue, yielding a further and consistent enhancement in performance.

### K.4  FULL RESULTS OF ABLATION STUDY OF DIFFERENT KEY-FREQUENCY PICKING STRATEGIES

Table 18 illustrates the complete results of the ablation study on various Key-Frequency Picking strategies. Notably, our **Softmax-based random sampling** strategy consistently achieves the best overall performance, particularly on more complex datasets.

Table 15: Multivariate long-term forecasting result comparison. We use prediction lengths $F \in \{96, 192, 336, 720\}$, and input length $T = 96$. The best results are in **bold**, and the second best are underlined.

| Model | | ReFocus (Ours) | | FilterNet (2024) | | iTransformer (2024b) | | ModernTCN (2024) | | FITS (2024b) | | PatchTST (2023) | | Crossformer (2023) | | TimesNet (2023) | | TSMixer (2023) | | DLinear (2023) | | FreTS (2023c) | |
|---|---|---|---|---|---|---|---|---|---|---|---|---|---|---|---|---|---|---|---|---|---|---|---|
| Metric | | MSE | MAE | MSE | MAE | MSE | MAE | MSE | MAE | MSE | MAE | MSE | MAE | MSE | MAE | MSE | MAE | MSE | MAE | MSE | MAE | MSE | MAE |
| ETTm1 | 96 | **0.321** | **0.360** | 0.321 | 0.361 | 0.334 | 0.368 | 0.317 | 0.362 | 0.355 | 0.375 | 0.329 | 0.367 | 0.404 | 0.426 | 0.338 | 0.375 | 0.323 | 0.363 | 0.345 | 0.372 | 0.335 | 0.372 |
| | 192 | **0.365** | **0.379** | 0.367 | 0.387 | 0.377 | 0.391 | 0.366 | 0.389 | 0.392 | 0.393 | 0.367 | 0.385 | 0.450 | 0.451 | 0.374 | 0.387 | 0.376 | 0.392 | 0.380 | 0.389 | 0.388 | 0.401 |
| | 336 | **0.398** | **0.400** | 0.401 | 0.409 | 0.426 | 0.420 | 0.407 | 0.412 | 0.424 | 0.414 | 0.399 | 0.410 | 0.532 | 0.515 | 0.410 | 0.411 | 0.407 | 0.413 | 0.413 | 0.413 | 0.421 | 0.426 |
| | 720 | 0.463 | **0.437** | 0.477 | 0.448 | 0.491 | 0.459 | 0.466 | 0.443 | 0.487 | 0.449 | **0.454** | 0.439 | 0.666 | 0.589 | 0.478 | 0.450 | 0.485 | 0.459 | 0.474 | 0.453 | 0.486 | 0.465 |
| | Avg | **0.387** | **0.394** | 0.392 | 0.401 | 0.407 | 0.410 | 0.389 | 0.402 | 0.415 | 0.408 | 0.387 | 0.400 | 0.513 | 0.496 | 0.400 | 0.406 | 0.398 | 0.407 | 0.403 | 0.407 | 0.408 | 0.416 |
| ETTm2 | 96 | **0.173** | **0.255** | 0.175 | 0.258 | 0.180 | 0.264 | 0.173 | 0.255 | 0.183 | 0.266 | 0.175 | 0.259 | 0.287 | 0.366 | 0.187 | 0.267 | 0.182 | 0.266 | 0.193 | 0.292 | 0.189 | 0.277 |
| | 192 | 0.237 | 0.297 | 0.240 | 0.301 | 0.250 | 0.309 | **0.235** | **0.296** | 0.247 | 0.305 | 0.241 | 0.302 | 0.414 | 0.492 | 0.249 | 0.309 | 0.249 | 0.309 | 0.284 | 0.362 | 0.258 | 0.326 |
| | 336 | **0.295** | **0.334** | 0.311 | 0.347 | 0.311 | 0.348 | 0.308 | 0.344 | 0.307 | 0.342 | 0.305 | 0.343 | 0.597 | 0.542 | 0.321 | 0.351 | 0.309 | 0.347 | 0.369 | 0.427 | 0.343 | 0.390 |
| | 720 | **0.395** | **0.392** | 0.414 | 0.405 | 0.412 | 0.407 | 0.398 | 0.394 | 0.407 | 0.399 | 0.402 | 0.400 | 1.730 | 1.042 | 0.408 | 0.403 | 0.416 | 0.408 | 0.554 | 0.522 | 0.495 | 0.480 |
| | Avg | **0.275** | **0.320** | 0.285 | 0.328 | 0.288 | 0.332 | 0.279 | 0.322 | 0.286 | 0.328 | 0.281 | 0.326 | 0.757 | 0.610 | 0.291 | 0.333 | 0.289 | 0.333 | 0.350 | 0.401 | 0.321 | 0.368 |
| ETTh1 | 96 | **0.376** | **0.394** | 0.382 | 0.402 | 0.386 | 0.405 | 0.386 | 0.394 | 0.386 | 0.396 | 0.414 | 0.419 | 0.423 | 0.448 | 0.384 | 0.402 | 0.401 | 0.412 | 0.386 | 0.400 | 0.395 | 0.407 |
| | 192 | **0.428** | **0.422** | 0.430 | 0.429 | 0.441 | 0.436 | 0.436 | 0.423 | 0.436 | 0.423 | 0.460 | 0.445 | 0.471 | 0.474 | 0.436 | 0.429 | 0.452 | 0.442 | 0.437 | 0.432 | 0.448 | 0.440 |
| | 336 | **0.462** | **0.442** | 0.472 | 0.451 | 0.487 | 0.458 | 0.479 | 0.445 | 0.478 | 0.444 | 0.501 | 0.466 | 0.570 | 0.546 | 0.491 | 0.469 | 0.492 | 0.463 | 0.481 | 0.459 | 0.499 | 0.472 |
| | 720 | **0.470** | **0.474** | 0.481 | 0.473 | 0.503 | 0.491 | 0.481 | 0.469 | 0.502 | 0.495 | 0.500 | 0.488 | 0.653 | 0.621 | 0.521 | 0.500 | 0.507 | 0.490 | 0.519 | 0.516 | 0.558 | 0.532 |
| | Avg | **0.434** | **0.433** | 0.441 | 0.439 | 0.454 | 0.447 | 0.446 | 0.433 | 0.451 | 0.440 | 0.469 | 0.454 | 0.529 | 0.522 | 0.458 | 0.450 | 0.463 | 0.452 | 0.456 | 0.452 | 0.475 | 0.463 |
| ETTh2 | 96 | **0.288** | **0.337** | 0.293 | 0.343 | 0.297 | 0.349 | 0.292 | 0.340 | 0.295 | 0.350 | 0.302 | 0.348 | 0.745 | 0.584 | 0.340 | 0.374 | 0.319 | 0.361 | 0.333 | 0.387 | 0.309 | 0.364 |
| | 192 | **0.371** | **0.390** | 0.374 | 0.396 | 0.380 | 0.400 | 0.377 | 0.395 | 0.381 | 0.396 | 0.388 | 0.400 | 0.877 | 0.656 | 0.402 | 0.414 | 0.402 | 0.410 | 0.477 | 0.476 | 0.395 | 0.425 |
| | 336 | **0.409** | **0.421** | 0.417 | 0.430 | 0.428 | 0.432 | 0.424 | 0.434 | 0.426 | 0.438 | 0.426 | 0.433 | 1.043 | 0.731 | 0.452 | 0.452 | 0.444 | 0.446 | 0.594 | 0.541 | 0.462 | 0.467 |
| | 720 | **0.417** | **0.436** | 0.449 | 0.460 | 0.427 | 0.445 | 0.433 | 0.448 | 0.431 | 0.446 | 0.431 | 0.446 | 1.104 | 0.763 | 0.462 | 0.468 | 0.441 | 0.450 | 0.831 | 0.657 | 0.721 | 0.604 |
| | Avg | **0.371** | **0.396** | 0.383 | 0.407 | 0.383 | 0.407 | 0.382 | 0.404 | 0.383 | 0.408 | 0.387 | 0.407 | 0.942 | 0.684 | 0.414 | 0.427 | 0.401 | 0.417 | 0.559 | 0.515 | 0.472 | 0.465 |
| ECL | 96 | **0.143** | **0.238** | 0.147 | 0.245 | 0.148 | 0.240 | 0.173 | 0.260 | 0.200 | 0.278 | 0.181 | 0.270 | 0.219 | 0.314 | 0.168 | 0.272 | 0.157 | 0.260 | 0.197 | 0.282 | 0.176 | 0.258 |
| | 192 | **0.158** | **0.252** | 0.160 | 0.250 | 0.162 | 0.253 | 0.181 | 0.267 | 0.200 | 0.280 | 0.188 | 0.274 | 0.231 | 0.322 | 0.184 | 0.289 | 0.173 | 0.274 | 0.196 | 0.285 | 0.175 | 0.262 |
| | 336 | **0.172** | **0.267** | 0.173 | 0.267 | 0.178 | 0.269 | 0.196 | 0.283 | 0.214 | 0.295 | 0.204 | 0.293 | 0.246 | 0.337 | 0.198 | 0.300 | 0.192 | 0.295 | 0.209 | 0.301 | 0.185 | 0.278 |
| | 720 | **0.198** | **0.290** | 0.210 | 0.309 | 0.225 | 0.317 | 0.238 | 0.316 | 0.255 | 0.327 | 0.246 | 0.324 | 0.280 | 0.363 | 0.220 | 0.320 | 0.223 | 0.318 | 0.245 | 0.333 | 0.220 | 0.315 |
| | Avg | **0.168** | **0.262** | 0.173 | 0.268 | 0.178 | 0.270 | 0.197 | 0.282 | 0.217 | 0.295 | 0.205 | 0.290 | 0.244 | 0.334 | 0.192 | 0.295 | 0.186 | 0.287 | 0.212 | 0.300 | 0.189 | 0.278 |
| Traffic | 96 | **0.380** | **0.248** | 0.430 | 0.294 | 0.395 | 0.268 | 0.550 | 0.355 | 0.651 | 0.391 | 0.462 | 0.295 | 0.522 | 0.290 | 0.593 | 0.321 | 0.493 | 0.336 | 0.650 | 0.396 | 0.593 | 0.378 |
| | 192 | **0.403** | **0.259** | 0.452 | 0.307 | 0.417 | 0.276 | 0.527 | 0.337 | 0.602 | 0.363 | 0.466 | 0.296 | 0.530 | 0.293 | 0.617 | 0.336 | 0.497 | 0.351 | 0.598 | 0.370 | 0.595 | 0.377 |
| | 336 | **0.419** | **0.267** | 0.470 | 0.316 | 0.433 | 0.283 | 0.537 | 0.342 | 0.609 | 0.366 | 0.482 | 0.304 | 0.558 | 0.305 | 0.629 | 0.336 | 0.528 | 0.361 | 0.605 | 0.373 | 0.609 | 0.385 |
| | 720 | **0.446** | **0.287** | 0.498 | 0.323 | 0.467 | 0.302 | 0.570 | 0.359 | 0.647 | 0.385 | 0.514 | 0.322 | 0.589 | 0.328 | 0.640 | 0.350 | 0.569 | 0.380 | 0.645 | 0.394 | 0.673 | 0.418 |
| | Avg | **0.412** | **0.265** | 0.463 | 0.310 | 0.428 | 0.282 | 0.546 | 0.348 | 0.627 | 0.376 | 0.481 | 0.304 | 0.550 | 0.304 | 0.620 | 0.336 | 0.522 | 0.357 | 0.625 | 0.383 | 0.618 | 0.390 |
| Weather | 96 | 0.160 | **0.202** | 0.162 | 0.207 | 0.174 | 0.214 | 0.165 | 0.203 | 0.166 | 0.213 | 0.177 | 0.218 | **0.158** | 0.230 | 0.172 | 0.220 | 0.166 | 0.210 | 0.196 | 0.255 | 0.174 | 0.208 |
| | 192 | 0.211 | 0.248 | 0.210 | 0.250 | 0.221 | 0.254 | 0.212 | **0.247** | 0.213 | 0.254 | 0.225 | 0.259 | **0.206** | 0.277 | 0.219 | 0.261 | 0.215 | 0.256 | 0.237 | 0.296 | 0.219 | 0.250 |
| | 336 | 0.266 | **0.290** | **0.265** | 0.290 | 0.278 | 0.296 | 0.266 | 0.293 | 0.269 | 0.294 | 0.278 | 0.297 | 0.272 | 0.335 | 0.280 | 0.306 | 0.287 | 0.300 | 0.283 | 0.335 | 0.273 | 0.290 |
| | 720 | 0.344 | 0.343 | **0.342** | **0.340** | 0.358 | 0.349 | 0.344 | 0.343 | 0.346 | 0.343 | 0.354 | 0.348 | 0.398 | 0.418 | 0.365 | 0.359 | 0.355 | 0.348 | 0.345 | 0.381 | 0.334 | 0.332 |
| | Avg | **0.245** | **0.271** | 0.245 | 0.272 | 0.258 | 0.279 | 0.247 | 0.272 | 0.249 | 0.276 | 0.259 | 0.281 | 0.259 | 0.315 | 0.259 | 0.287 | 0.256 | 0.279 | 0.265 | 0.317 | 0.250 | 0.270 |
| Solar_Energy | 96 | **0.182** | **0.219** | 0.206 | 0.251 | 0.203 | 0.237 | 0.206 | 0.264 | 0.371 | 0.417 | 0.234 | 0.286 | 0.310 | 0.331 | 0.250 | 0.292 | 0.221 | 0.275 | 0.290 | 0.378 | 0.217 | 0.278 |
| | 192 | **0.222** | **0.249** | 0.242 | 0.279 | 0.233 | 0.261 | 0.246 | 0.285 | 0.377 | 0.398 | 0.267 | 0.310 | 0.734 | 0.725 | 0.296 | 0.318 | 0.268 | 0.306 | 0.320 | 0.398 | 0.256 | 0.302 |
| | 336 | **0.240** | **0.268** | 0.255 | 0.291 | 0.248 | 0.273 | 0.260 | 0.296 | 0.416 | 0.412 | 0.290 | 0.315 | 0.750 | 0.735 | 0.319 | 0.330 | 0.272 | 0.294 | 0.353 | 0.415 | 0.263 | 0.307 |
| | 720 | **0.242** | **0.271** | 0.267 | 0.301 | 0.249 | 0.275 | 0.264 | 0.298 | 0.414 | 0.400 | 0.289 | 0.317 | 0.769 | 0.765 | 0.338 | 0.337 | 0.281 | 0.313 | 0.356 | 0.413 | 0.256 | 0.297 |
| | Avg | **0.222** | **0.252** | 0.243 | 0.283 | 0.233 | 0.262 | 0.244 | 0.286 | 0.395 | 0.407 | 0.270 | 0.307 | 0.641 | 0.639 | 0.301 | 0.319 | 0.260 | 0.297 | 0.330 | 0.401 | 0.248 | 0.296 |
| $1^{st}$ Count | | 34 | 36 | 2 | 2 | 0 | 0 | 1 | 2 | 0 | 0 | 1 | 0 | 2 | 0 | 0 | 0 | 0 | 0 | 0 | 0 | 0 | 0 |

Table 16: Full result of ablation study on the 'Adaptive Mid-Frequency Energy Optimizer (AMEO)' and the 'Key-Frequency Enhanced Training (KET)' strategy. We use prediction lengths $F \in \{96, 192, 336, 720\}$, and input length $T = 96$. The best results are in **bold**.

| Model | | Both (ReFocus) MSE | MAE | + AMEO MSE | MAE | + KET MSE | MAE | None MSE | MAE |
|---|---|---|---|---|---|---|---|---|---|
| ETTm1 | 96 | **0.321** | **0.360** | 0.331 | 0.368 | 0.331 | 0.363 | 0.339 | 0.367 |
| | 192 | **0.365** | **0.379** | 0.377 | 0.390 | 0.373 | 0.382 | 0.381 | 0.391 |
| | 336 | **0.398** | **0.400** | 0.403 | 0.407 | 0.403 | 0.402 | 0.414 | 0.413 |
| | 720 | 0.463 | **0.437** | **0.462** | 0.441 | 0.467 | 0.438 | 0.468 | 0.442 |
| | Avg | **0.387** | **0.394** | 0.393 | 0.402 | 0.394 | 0.396 | 0.401 | 0.403 |
| ETTm2 | 96 | **0.173** | **0.255** | 0.179 | 0.262 | 0.178 | 0.260 | 0.180 | 0.262 |
| | 192 | **0.237** | **0.297** | 0.244 | 0.304 | 0.241 | 0.299 | 0.245 | 0.302 |
| | 336 | **0.295** | **0.334** | 0.304 | 0.340 | 0.300 | 0.337 | 0.304 | 0.340 |
| | 720 | **0.395** | **0.392** | 0.402 | 0.396 | 0.398 | 0.393 | 0.404 | 0.396 |
| | Avg | **0.275** | **0.320** | 0.282 | 0.326 | 0.279 | 0.322 | 0.283 | 0.325 |
| ETTh1 | 96 | **0.376** | **0.394** | 0.382 | 0.398 | 0.378 | 0.395 | 0.383 | 0.395 |
| | 192 | **0.428** | **0.422** | 0.433 | 0.425 | 0.432 | 0.423 | 0.432 | 0.425 |
| | 336 | **0.462** | **0.442** | 0.468 | 0.450 | 0.469 | 0.447 | 0.469 | 0.449 |
| | 720 | **0.470** | **0.474** | 0.489 | 0.486 | 0.470 | 0.474 | 0.474 | 0.480 |
| | Avg | **0.434** | **0.433** | 0.443 | 0.440 | 0.437 | 0.435 | 0.440 | 0.437 |
| ETTh2 | 96 | 0.288 | **0.337** | **0.285** | 0.336 | 0.289 | 0.339 | 0.288 | 0.338 |
| | 192 | **0.371** | **0.390** | 0.375 | 0.391 | 0.374 | 0.390 | 0.374 | 0.391 |
| | 336 | **0.409** | **0.421** | 0.405 | 0.420 | 0.412 | 0.425 | 0.419 | 0.428 |
| | 720 | **0.417** | **0.436** | 0.424 | 0.441 | 0.418 | 0.438 | 0.423 | 0.441 |
| | Avg | **0.371** | **0.396** | 0.372 | 0.397 | 0.373 | 0.398 | 0.376 | 0.400 |
| ECL | 96 | **0.143** | **0.238** | 0.146 | 0.241 | 0.145 | 0.239 | 0.147 | 0.242 |
| | 192 | **0.158** | **0.252** | 0.165 | 0.259 | 0.161 | 0.253 | 0.162 | 0.256 |
| | 336 | **0.172** | **0.267** | 0.177 | 0.272 | 0.176 | 0.269 | 0.180 | 0.274 |
| | 720 | **0.198** | **0.290** | 0.206 | 0.297 | 0.203 | 0.292 | 0.221 | 0.307 |
| | Avg | **0.168** | **0.262** | 0.174 | 0.267 | 0.171 | 0.263 | 0.178 | 0.270 |
| Traffic | 96 | **0.380** | **0.248** | 0.414 | 0.274 | 0.380 | 0.250 | 0.414 | 0.278 |
| | 192 | **0.403** | **0.259** | 0.439 | 0.287 | 0.404 | 0.262 | 0.437 | 0.284 |
| | 336 | **0.419** | **0.267** | 0.449 | 0.288 | 0.421 | 0.270 | 0.449 | 0.288 |
| | 720 | **0.446** | **0.287** | 0.506 | 0.307 | 0.450 | 0.290 | 0.495 | 0.307 |
| | Avg | **0.412** | **0.265** | 0.452 | 0.289 | 0.414 | 0.268 | 0.449 | 0.289 |
| Weather | 96 | **0.160** | **0.202** | 0.165 | 0.209 | 0.164 | 0.207 | 0.164 | 0.209 |
| | 192 | 0.211 | **0.248** | **0.210** | 0.252 | 0.215 | 0.252 | 0.216 | 0.256 |
| | 336 | **0.266** | **0.290** | 0.267 | 0.291 | 0.273 | 0.295 | 0.275 | 0.299 |
| | 720 | **0.344** | **0.343** | 0.350 | 0.346 | 0.349 | 0.345 | 0.353 | 0.349 |
| | Avg | **0.245** | **0.271** | 0.248 | 0.275 | 0.250 | 0.275 | 0.252 | 0.278 |
| Solar_Energy | 96 | **0.182** | **0.219** | 0.197 | 0.226 | 0.192 | 0.230 | 0.192 | 0.234 |
| | 192 | **0.222** | **0.249** | 0.236 | 0.269 | 0.231 | 0.255 | 0.235 | 0.265 |
| | 336 | **0.240** | **0.268** | 0.246 | 0.276 | 0.244 | 0.271 | 0.249 | 0.279 |
| | 720 | **0.242** | **0.271** | 0.245 | 0.274 | 0.245 | 0.274 | 0.250 | 0.278 |
| | Avg | **0.222** | **0.252** | 0.231 | 0.261 | 0.228 | 0.258 | 0.232 | 0.264 |

Table 17: Full result of further ablation study on the 'Key-Frequency Enhanced Training (KET)' strategy. We use prediction lengths $F \in \{96, 192, 336, 720\}$, and input length $T = 96$. The best results are in **bold**.

| Model | | Both (KET) | | Pseudo | | Real | |
|---|---|---|---|---|---|---|---|
| Metric | | MSE | MAE | MSE | MAE | MSE | MAE |
| ETTm1 | 96 | 0.331 | 0.363 | **0.331** | **0.362** | 0.339 | 0.367 |
| | 192 | **0.373** | **0.382** | 0.375 | 0.384 | 0.381 | 0.391 |
| | 336 | **0.403** | **0.402** | 0.406 | 0.405 | 0.414 | 0.413 |
| | 720 | **0.467** | **0.438** | 0.471 | 0.440 | 0.468 | 0.442 |
| | Avg | **0.394** | **0.396** | 0.396 | 0.398 | 0.401 | 0.403 |
| ETTm2 | 96 | **0.178** | **0.260** | 0.178 | 0.260 | 0.180 | 0.262 |
| | 192 | **0.241** | **0.299** | 0.242 | 0.299 | 0.245 | 0.302 |
| | 336 | **0.300** | **0.337** | 0.301 | 0.339 | 0.304 | 0.340 |
| | 720 | **0.398** | **0.393** | 0.399 | 0.393 | 0.404 | 0.396 |
| | Avg | **0.279** | **0.322** | 0.280 | 0.323 | 0.283 | 0.325 |
| ETTh1 | 96 | **0.378** | **0.395** | 0.382 | 0.394 | 0.383 | 0.395 |
| | 192 | 0.432 | 0.423 | **0.429** | **0.423** | 0.432 | 0.425 |
| | 336 | 0.469 | 0.447 | **0.467** | **0.445** | 0.469 | 0.449 |
| | 720 | 0.470 | 0.474 | **0.467** | **0.472** | 0.474 | 0.480 |
| | Avg | 0.437 | 0.435 | **0.436** | **0.434** | 0.440 | 0.437 |
| ETTh2 | 96 | 0.289 | 0.339 | **0.288** | **0.338** | 0.288 | 0.338 |
| | 192 | 0.374 | 0.390 | **0.370** | **0.390** | 0.374 | 0.391 |
| | 336 | 0.412 | 0.425 | **0.412** | **0.423** | 0.419 | 0.428 |
| | 720 | 0.418 | 0.438 | **0.418** | **0.437** | 0.423 | 0.441 |
| | Avg | 0.373 | 0.398 | **0.372** | **0.397** | 0.376 | 0.400 |
| ECL | 96 | **0.145** | **0.239** | 0.147 | 0.241 | 0.147 | 0.242 |
| | 192 | **0.161** | **0.253** | 0.165 | 0.257 | 0.162 | 0.256 |
| | 336 | **0.176** | **0.269** | 0.179 | 0.271 | 0.180 | 0.274 |
| | 720 | **0.203** | **0.292** | 0.209 | 0.296 | 0.221 | 0.307 |
| | Avg | **0.171** | **0.263** | 0.175 | 0.266 | 0.178 | 0.270 |
| Traffic | 96 | **0.380** | **0.250** | 0.383 | 0.254 | 0.414 | 0.278 |
| | 192 | **0.404** | **0.262** | 0.406 | 0.265 | 0.437 | 0.284 |
| | 336 | **0.421** | **0.270** | 0.424 | 0.272 | 0.449 | 0.288 |
| | 720 | **0.450** | **0.290** | 0.454 | 0.293 | 0.495 | 0.307 |
| | Avg | **0.414** | **0.268** | 0.417 | 0.271 | 0.449 | 0.289 |
| Weather | 96 | **0.164** | **0.207** | 0.166 | 0.207 | 0.164 | 0.209 |
| | 192 | **0.215** | **0.252** | 0.216 | 0.255 | 0.216 | 0.256 |
| | 336 | **0.273** | **0.295** | 0.275 | 0.297 | 0.275 | 0.299 |
| | 720 | **0.349** | **0.345** | 0.352 | 0.346 | 0.353 | 0.347 |
| | Avg | **0.250** | **0.275** | 0.253 | 0.276 | 0.252 | 0.278 |
| Solar_Energy | 96 | **0.192** | **0.230** | 0.235 | 0.263 | 0.192 | 0.234 |
| | 192 | **0.231** | **0.255** | 0.290 | 0.303 | 0.235 | 0.265 |
| | 336 | **0.244** | **0.271** | 0.287 | 0.301 | 0.249 | 0.279 |
| | 720 | **0.245** | **0.274** | 0.296 | 0.308 | 0.250 | 0.278 |
| | Avg | **0.228** | **0.258** | 0.277 | 0.294 | 0.232 | 0.264 |

Table 18: Full result about ablation study of different Key-Frequency Picking strategies. We use prediction lengths $F \in \{96, 192, 336, 720\}$, and input length $T = 96$. The best results are in **bold**.

| Model | | **Softmax** | | Max | | Min | |
|---|---|---|---|---|---|---|---|
| Metric | | MSE | MAE | MSE | MAE | MSE | MAE |
| ETTm1 | 96 | 0.321 | 0.360 | 0.331 | 0.360 | **0.321** | **0.357** |
| | 192 | **0.365** | 0.379 | 0.370 | 0.380 | 0.366 | **0.377** |
| | 336 | **0.398** | 0.400 | 0.401 | 0.402 | 0.400 | **0.399** |
| | 720 | **0.463** | 0.437 | 0.467 | 0.438 | 0.464 | **0.436** |
| | Avg | **0.387** | 0.394 | 0.392 | 0.395 | 0.388 | **0.392** |
| ETTm2 | 96 | **0.173** | **0.255** | 0.175 | 0.258 | 0.177 | 0.259 |
| | 192 | **0.237** | **0.297** | 0.240 | 0.300 | 0.242 | 0.300 |
| | 336 | **0.295** | **0.334** | 0.303 | 0.338 | 0.302 | 0.338 |
| | 720 | **0.395** | **0.392** | 0.396 | 0.392 | 0.398 | 0.394 |
| | Avg | **0.275** | **0.320** | 0.279 | 0.322 | 0.280 | 0.323 |
| ETTh1 | 96 | 0.376 | 0.394 | 0.380 | 0.396 | **0.372** | **0.391** |
| | 192 | 0.428 | **0.422** | 0.430 | 0.423 | **0.426** | 0.423 |
| | 336 | **0.462** | **0.442** | 0.464 | 0.444 | 0.464 | 0.443 |
| | 720 | 0.470 | 0.474 | 0.473 | 0.478 | **0.467** | **0.472** |
| | Avg | 0.434 | 0.433 | 0.437 | 0.435 | **0.432** | **0.432** |
| ETTh2 | 96 | 0.288 | 0.337 | 0.289 | 0.340 | **0.287** | **0.337** |
| | 192 | 0.371 | 0.390 | 0.373 | 0.391 | **0.366** | **0.388** |
| | 336 | **0.409** | **0.421** | 0.414 | 0.425 | 0.410 | 0.423 |
| | 720 | **0.417** | **0.436** | 0.418 | 0.437 | 0.419 | 0.437 |
| | Avg | 0.371 | **0.396** | 0.374 | 0.398 | **0.371** | 0.396 |
| ECL | 96 | **0.143** | **0.238** | 0.145 | 0.241 | 0.165 | 0.256 |
| | 192 | **0.158** | **0.252** | 0.162 | 0.256 | 0.176 | 0.267 |
| | 336 | **0.172** | **0.267** | 0.175 | 0.269 | 0.192 | 0.283 |
| | 720 | **0.198** | **0.290** | 0.204 | 0.293 | 0.242 | 0.318 |
| | Avg | **0.168** | **0.262** | 0.172 | 0.265 | 0.194 | 0.281 |
| Traffic | 96 | **0.380** | **0.248** | 0.389 | 0.253 | 0.504 | 0.341 |
| | 192 | **0.403** | **0.259** | 0.413 | 0.268 | 0.505 | 0.338 |
| | 336 | **0.419** | **0.267** | 0.427 | 0.276 | 0.521 | 0.351 |
| | 720 | **0.446** | **0.287** | 0.457 | 0.296 | 0.536 | 0.347 |
| | Avg | **0.412** | **0.265** | 0.422 | 0.273 | 0.517 | 0.344 |
| Weather | 96 | **0.160** | **0.202** | 0.166 | 0.207 | 0.164 | 0.205 |
| | 192 | **0.211** | **0.248** | 0.212 | 0.248 | 0.212 | 0.249 |
| | 336 | **0.266** | **0.290** | 0.268 | 0.291 | 0.269 | 0.290 |
| | 720 | **0.344** | **0.343** | 0.348 | 0.344 | 0.349 | 0.344 |
| | Avg | **0.245** | **0.271** | 0.249 | 0.273 | 0.249 | 0.272 |
| Solar_Energy | 96 | **0.182** | **0.219** | 0.189 | 0.228 | 0.208 | 0.247 |
| | 192 | **0.222** | **0.249** | 0.236 | 0.262 | 0.242 | 0.270 |
| | 336 | **0.240** | **0.268** | 0.245 | 0.273 | 0.256 | 0.283 |
| | 720 | **0.242** | **0.271** | 0.248 | 0.276 | 0.253 | 0.280 |
| | Avg | **0.222** | **0.252** | 0.230 | 0.260 | 0.240 | 0.270 |

## L   LIMITATIONS AND FUTURE WORKS

**Limitations**   Despite ReFocus demonstrating significant improvements in multivariate time series forecasting by addressing the mid-frequency spectrum gap and exploiting shared key-frequency information, several limitations remain. First, our experiments primarily rely on benchmark datasets (Traffic, ECL, Solar), which may not capture the full variability of more diverse, high-frequency, or non-stationary applications. Additionally, further analysis is needed to understand the robustness of our approach under abrupt distribution shifts or extreme noise conditions.

**Future Works**   Future work should extend the ReFocus framework to a broader range of application domains, such as financial, medical, and human activity time series, to verify its generalizability. We plan to explore adaptive mechanisms to dynamically tune module parameters and mitigate sensitivity issues. Moreover, investigating lightweight or hybrid architectures could improve real-time forecasting efficiency. Incorporating external contextual information may further enrich inter-channel dependency modeling. All these will help refine the overall design and improve resilience against challenging, non-stationary environments.

## M   SOCIETAL IMPACTS

The development of the ReFocus forecaster has the potential to significantly benefit various fields, such as finance and traffic, by improving the accuracy and efficiency of time series forecasting, thereby enhancing decision-making processes. However, there are potential negative societal impacts to consider. Privacy concerns may arise from the use of personal data, especially in healthcare and finance, leading to possible violations. Additionally, biases in the data could result in unfair outcomes, perpetuating existing disparities. Over-reliance on automated forecasting models might lead to the neglect of important context, causing adverse outcomes. To mitigate these risks, robust data protection protocols should be implemented, and continuous monitoring for bias is necessary to ensure fairness. Developing ethical use policies and maintaining human oversight in decision-making can further ensure that the deployment of ReFocus maximizes its positive societal impact while minimizing potential negative consequences.

