# OpenReview forum: "Bridging the Spectrum Gap: Mid‑Frequency Augmentation and Key‑Frequency Mining for Multivariate Time Series"
_ICLR.cc/2026/Conference — ICLR 2026 Conference Withdrawn Submission_

### Official Review · Reviewer_JeA2 · 2025-10-29

**Soundness:** 2
**Presentation:** 3
**Contribution:** 2
**Rating:** 2
**Confidence:** 4

**Summary:**

This paper addresses two underexplored challenges in frequency-based time series forecasting: the Mid-Frequency Spectrum Gap and the Shared Key-Frequency phenomenon across correlated variables. The authors propose three components: an Adaptive Mid-Frequency Energy Optimizer, an Energy-based Key-Frequency Picking Block, and a Key-Frequency Enhanced Training strategy, to emphasize mid-frequency information and exploit shared spectral structures among multiple series. Experiments on Traffic, Electricity, and Solar benchmarks demonstrate consistent improvements over strong baselines.

**Strengths:**

1. The paper is clearly written and easy to follow.
2. Experimental results are reported on multiple public datasets with standard baselines.

**Weaknesses:**

1. The motivation “the middle-frequency band has much lower energy than the low-frequency band, leading to an information utilization bottleneck” has two issues:

* First, the paper does not provide any experimental evidence to verify whether low-energy components are indeed useful for forecasting tasks. In the time series forecasting literature, some studies (e.g., FITS) intentionally truncate the spectrum and discard low-energy information. If low-energy components are not beneficial to prediction, this motivation would not hold.

* Second, this motivation is not novel. Similar arguments have already been discussed in prior works such as Amplifier: Bringing Attention to Neglected Low-Energy Components in Time Series Forecasting (AAAI 2025) and Fredformer: Frequency Debiased Transformer for Time Series Forecasting (KDD 2024).

2. In Section 2 RELATED WORK, the authors divide existing time series forecasting models into three categories:
(1) the application of sequential models to time series data,
(2) the tokenization of time series, and
(3) the exploration of intrinsic patterns within time series.

- However, this categorization appears uncommon and somewhat unclear in its intention. It is not evident what specific conceptual or methodological distinction the authors aim to emphasize through this classification — whether it is based on modeling paradigms, data representations, or learning objectives.
Moreover, the classification itself seems inaccurate. For example, the authors classify Graph Neural Networks (GNNs) under “(1) the application of sequential models to time series data,” whereas GNNs are not sequential models.
It would be helpful for the authors to clarify the rationale behind this taxonomy and to reconsider the categorization to better reflect the methodological differences among existing approaches.

3. Criticizing RevIN for not addressing the Mid-Frequency Spectrum Gap seems somewhat misleading or irrelevant. The primary objective of RevIN is to mitigate non-stationarity in time series data through instance-wise normalization and rescaling, and addressing the Mid-Frequency Spectrum Gap has never been within its design scope.

4. The definition of Definition 3.1 (Frequency Spectral Energy) lacks proper citations to classical references on the Fourier Transform and related spectral analysis literature.

5. In Figure 2, the specific operations of AMEO are not shown. Many details in the Energy-based Key-Frequency Picking Block are confusing, for example, what does $H_i^k∈R^{\(C×Q\)}$ represent, and how is it obtained? What do the red and black numbers in the matrix $H_i^f∈R^{(C×(Q/2+1))}$ mean?

6. Regarding Definition 3.4 (Adaptive Mid-Frequency Energy Optimizer), the author states that Equation (4) is equivalent to $x=x-β⋅Conv⁡(x)$. However, since this formulation is similar to the 1D convolution used in the EncoderLayer of transformer-based models, I do not consider it to be novel.

7. The baselines selected in this paper are not sufficiently up-to-date. The most recent baseline, FilterNet (NeurIPS 2024), was released a year ago. The work lacks comparisons with models published in ICML 2025 and ICLR 2025. Moreover, based on the specific performance values of ReFocus reported in Table 15, I would say that, at this point, such performance does not reach the state-of-the-art (SOTA) level.

8. Line 101: The phrase “To address challenge 2, for the second challenge,” is redundant.

**Questions:**

1. I would like to know whether the original data without using RevIN already exhibits an imbalanced energy distribution.

2. In the sentence “We propose KET, where spectral information from other channels is randomly introduced into each channel, to enhance the extraction of the shared Key-Frequency,” why is the introduction *random*? Is this choice reasonable?

---

### Official Review · Reviewer_CjZL · 2025-10-31

**Soundness:** 3
**Presentation:** 3
**Contribution:** 3
**Rating:** 6
**Confidence:** 4

**Summary:**

This paper addresses two significant challenges in multivariate time series analysis: the Mid-Frequency Spectrum Gap—where spectral energy in real-world signals is underrepresented in the mid-frequencies—and the modeling of shared key-frequency components across variables. The authors propose ReFocus, a new framework featuring (i) the Adaptive Mid-Frequency Energy Optimizer (AMEO) to boost mid-frequency information, (ii) the Energy-based Key-Frequency Picking Block (EKPB) for efficient cross-series spectral modeling, and (iii) a Key-Frequency Enhanced Training (KET) strategy that augment training via inter-channel spectral mixup. Experiments on eight benchmarks demonstrate strong empirical gains, challenging the leading iTransformer and other recent baselines.

**Strengths:**

1. Strong Problem Motivation and Theoretical Rigor: The paper successfully identifies and tackles two critical, well-motivated challenges in multivariate time series forecasting: the Mid-Frequency Spectrum Gap (MFSG) and the efficient modeling of shared frequency patterns. The problem-solving approach is backed by rigorous theoretical analysis (e.g., Theorems 3.3 and 3.5), which clearly explains the limitations of existing baselines (RevIN, standard filters) and mathematically justifies the effectiveness of the proposed Adaptive Mid-Frequency Energy Optimizer (AMEO) in stabilizing the spectrum.

2. Innovative and Parameter-Efficient Architecture: The core framework components—AMEO, Energy-based Key-Frequency Picking Block (EKPB), and Key-Frequency Enhanced Training (KET)—are highly novel and strategically designed.

3. Comprehensive and Deep Empirical Validation: The experimental section is exceptionally thorough. ReFocus consistently achieves new State-of-the-Art (SOTA) results across a wide range of eight large-scale multivariate datasets (Table 15), showing particularly significant and substantial gains on complex, high-channel-count benchmarks (Traffic, ECL, Solar).

**Weaknesses:**

1. While Theorem 3.5 clearly shows how AMEO reshapes the spectrum, the theoretical treatment lacks a deeper analysis regarding the statistical properties of the mid-frequency enhancement. Specifically, there is no discussion on:

>Whether AMEO risks amplifying irrelevant noise or non-signal structure present in the middle frequency bands.
>Formal limits or bounds on the enhancement factor $\beta$. Is there a risk of "overfilling" the spectrum, and if so, what are the empirical or theoretical safeguards?

2. The core mechanism of the Energy-based Key-Frequency Picking Block (EKPB) relies on detecting shared spectral energy and correlation (Figure 1, Figure 5). The paper does not address the fundamental challenge in multivariate time series of distinguishing causal cross-channel structure from confounded/non-causal correlation driven by latent common factors. Relying purely on energy similarity may lead to suboptimal feature picking in complex, real-world systems.

**Questions:**

1. Scope of the Mid-Frequency Gap: Have the authors analyzed the “mid-frequency gap” on naturally nonstationary or multiresolution datasets outside the current forecasting suite? Are there circumstances where mid-frequency amplification could amplify noise (e.g., data with rapid spikes or heavy-tailed distributions)?
2. Have you tested ReFocus on real-world time series with strong hierarchical or causal structure, such as supply chain or sensor networks? How robust is the approach to missing data or irregular sampling schedules?
3. Could the authors add direct empirical comparison to leading frequency-domain or time-domain data augmentation approaches to contextualize the relative benefits and generality of KET?
4. Have the authors investigated scenarios or domains where mid-frequency boosting harms predictive performance? For example, in domains where crucial structure is concentrated in low or high frequencies and the mid-band is genuinely data-sparse?
5. The AMEO module removes low-frequency trends by subtracting a local mean (the result of a low-pass filter), which is essentially a high-pass filtering operation. Does this "destructive" detrending operation not risk discarding critical predictive information, particularly for long-term forecasting tasks? In the final stage of prediction, how does the model recover or compensate for this removed trend information?
6. The core of the method is to enhance mid-frequency components. However, in many real-world signals, the mid-frequency band may contain not only meaningful periodic patterns but also significant pseudo-periodic noise or irrelevant interference. While amplifying the signal, how does AMEO distinguish between meaningful patterns and spurious noise? Is there a mechanism in place to prevent the model from overfitting to these amplified noise patterns, especially on datasets with a low Signal-to-Noise Ratio (SNR)?
7. The success of AMEO seems to rely on a core assumption: that the low-frequency trend causing non-stationarity is separable from the mid-frequency patterns that hold predictive value. In certain time series (e.g., economic cycles), the trend itself may exhibit periodic behavior, or predictive information might be entangled across both low and mid frequencies. In such cases, does AMEO risk erroneously removing critical signal components? What are the boundary conditions for the effectiveness of this method?

---

> ### Author Response · Authors · 2025-11-19
> **Genuine Appreciation**
>
> Dear Reviewer CjZL,
>
> Thank you so much for your thoughtful and encouraging review. We truly appreciate the time you spent evaluating our work and the constructive insights you provided. They were genuinely helpful, and we’re grateful for the recognition you gave to our motivation, analysis, and experimental effort.
>
> We’re very sorry that we won’t be able to continue the discussion, as we’ve decided to withdraw the submission. Still, your comments will definitely guide how we improve the work moving forward.
>
> Thanks again for your professionalism and kindness throughout the process.
>
> ***In case I don’t see you, good afternoon, good evening, and good night.*** Wishing you all the best ahead！

---

### Official Review · Reviewer_PhzE · 2025-11-01

**Soundness:** 2
**Presentation:** 3
**Contribution:** 2
**Rating:** 4
**Confidence:** 4

**Summary:**

This paper proposes Residual Frequency Optimization and Cross-channel Unified Spectrum modeling, and introduce a unified framework, ReFocus, to address two issues: the mid-frequency spectrum gap and shared key-frequency modeling.

**Strengths:**

1. This paper demonstrates, through both theoretical analysis and empirical evidence, that existing approaches such as RevIN and conventional high and low-pass filters fail to resolve the mid-frequency spectrum gap.
2. This paper introduces EKPB and KET to capture and enhance cross-channel shared Key-Frequencies, achieving superior inter-series modeling with fewer parameters by randomly injecting spectral information across channels.
3. Extensive long-term forecasting and visualization experiments validate the effectiveness of the proposed ReFocus.

**Weaknesses:**

1. Several claims in the Introduction are insufficiently supported. In particular, the statement that "Mid-Frequency Spectrum Gap will introduce Nonstationarity" lacks appropriate evidence, and the cited non-stationary Transformer work does not adequately substantiate this assertion.
2. The authors argue that high- and low-pass filters and RevIN cannot address the mid-frequency spectrum gap, providing theoretical analysis in Section 3.2 and empirical evidence in Figure 4. However, it remains unclear whether the learnable filters used by existing forecasting models such as FilterNet [1] and TSLANet [2] can resolve this issue. Can TimeKAN [3], which adopts a frequency-decomposed learning paradigm, address this issue? A correspondingly comprehensive theoretical and empirical analysis should be provided.
3. The paper should include efficiency evaluations against more lightweight models (e.g., FilterNet), reporting both parameter size and computational time.

[1] FilterNet: Harnessing Frequency Filters for Time Series Forecasting. NeurIPS, 2024.
[2] TSLANet: Rethinking Transformers for Time Series Representation Learning. ICML, 2024.
[3] TimeKAN: KAN-based Frequency Decomposition Learning Architecture for Long-term Time Series Forecasting. ICLR, 2025.

**Questions:**

pls refer to weakness.

---

### Official Review · Reviewer_78RT · 2025-11-02

**Soundness:** 2
**Presentation:** 3
**Contribution:** 2
**Rating:** 4
**Confidence:** 4

**Summary:**

This paper addresses two key challenges in multivariate time series forecasting: (1) the prevalent "Mid-Frequency Spectrum Gap" in real-world time series, where energy concentrated at low frequencies leads to the neglect of crucial mid-frequency information; and (2) the inefficient modeling of "Shared Key-Frequencies" across different channels in multivariate data. The empirical results show that ReFocus outperforms previous SOTA models (like iTransformer) on multiple benchmark datasets (especially Traffic, ECL, Solar) while maintaining low computational complexity (linear complexity).

**Strengths:**

1. SOTA Empirical Results: The model achieves SOTA performance on multiple challenging long-term time series forecasting benchmarks, particularly on datasets with a high number of variables (e.g., Traffic, ECL, Solar).

1. Efficient Module Design: The AMEO and EKPB modules are computationally efficient. EKPB, in particular, uses energy-based weighted averaging to model inter-series dependencies with linear complexity O(N), which is highly beneficial for high-dimensional time series.

1. Good Attempt at Interpretability: The paper uses visualizations  to help explain the modules' mechanisms, enhancing the model's credibility.

**Weaknesses:**

## W1: Vague and Contradictory Definition of the "Mid-Frequency Gap."


Vague Definition: The concept of the "Mid-Frequency Spectrum Gap" is the foundational premise for the AMEO module. However, the paper fails to establish quantitative boundaries for this 'mid-frequency' band. Without defining a specific frequency range (e.g., $f_{low} < f < f_{high}$) derived from the data properties, the "gap" remains a purely subjective, visual interpretation of Figure 1. This omission is critical because it makes it impossible to objectively verify if the AMEO module is selectively enhancing a specific, identified frequency band, as claimed, or merely acting as a generic high-pass filter across a broad, undefined spectrum. The lack of a rigorous, quantifiable definition undermines the scientific claim that the model solves a precisely identified problem.

Core Contradiction: The paper explicitly argues against the use of simple High-Pass Filters (HPFs), claiming they are ineffective or even detrimental (citing the poor result for HPF in Figure 4). However, the proposed AMEO module is defined by a residual connection: $\mathbf{X}' = \mathbf{X} - \beta \cdot \text{LowPassFilter}(\mathbf{X})$, where the low-pass filter is implemented via a depth-wise convolution. By definition, subtracting a low-pass component from the original signal results in a residual high-pass filter (RHPF). This creates a severe internal contradiction: the solution proposed (AMEO/RHPF) is mechanistically equivalent to the class of methods the paper dismisses (HPF). The authors fail to rigorously explain why their specific trainable, residual form of high-pass filtering succeeds, while the generalized HPF fails, strongly suggesting the paper employs a straw man argument against simple filtering to artificially elevate the AMEO's novelty.
## W2: Overly Strong Inductive Bias in EKPB Module.

The EKPB module computes a single, weighted-average "shared key-frequency" blueprint and applies this identical blueprint back to all channels.

This "one-size-fits-all" assumption that all channels share a single, dominant frequency pattern might be effective for homogeneous datasets (e.g., power consumption), but it is questionable for highly heterogeneous datasets (e.g., Weather, with 21 distinct physical variables).



## W3: Insufficient Comparison to Recent State-of-the-Art (SOTA) Baselines.

The paper does not include critical recent baselines in the frequency/decomposition domain, such as FreDF and CycleNet, or other strong contemporary models. The omission of direct comparison against the most recent SOTA weakens the empirical claim and makes it impossible to fully assess the model's true effectiveness.

**Questions:**

See weaknesses

---

### Note · Authors · 2025-11-19

I have read and agree with the venue's withdrawal policy on behalf of myself and my co-authors.